# Emergence of alternative states in a synthetic human gut microbial community

**Daniel Rios Garza** [1,2,7] ✉, **Bin Liu** [1,3,4,7], **Charlotte van de Velde** [1,7], **Xingjian Zhou** [1], **Pallabita Saha** [1], **Didier Gonze** [5], **Kenneth Simoens** [6], **Kristel Bernaerts** [6] & **Karoline Faust** [1] ✉

Several human-associated microbial communities exist in multiple configurations and can change their composition in response to perturbations, remaining in an altered state even after the perturbation ends. Multistability has been previously proposed to explain this behavior for gut microbiota in particular, but has not been clearly demonstrated experimentally. Here, we first investigate the life history strategies of three common human gut bacteria to identify mechanisms driving alternative states. We then use this data to build and parameterize a kinetic model, which predicts that alternative states emerge due to phenotype switching between subpopulations of the same species. Perturbation experiments support these predictions, and confirm the existence of alternative states. Finally, simulations show that phenotype switching can also explain alternative states in larger communities. Thus, a transient perturbation combined with metabolic flexibility is sufficient for alternative communities to emerge.

Several human-associated microbial communities assemble into more than one configuration[1–3] and change their composition in response to perturbations, remaining in an altered state even after the perturbation ends[4,5]. While various hypotheses have been proposed to explain this behavior[6–8], clear demonstrations of the mechanisms underlying these hypotheses are still lacking. One common explanation, supported by empirical evidence, suggests that microbiomes, like many ecological systems, can assemble into alternative stable states[7]. Thus, even when the same microbes are assembled under similar environmental conditions, they may converge on distinct community compositions, influenced by their assembly history[9]. Additionally, minor yet continuous changes in environmental parameters could lead to significant shifts in community states[10]. Mechanistically identifying alternative stable states and regime shifts in natural communities has historically been

challenging[11]. This is because organisms can change their own environments—microbes, for example, consume resources[12], change the environment's pH[13], and create physical structures[14]—in response to biotic and abiotic changes, thereby making it difficult to disentangle community states from environmental factors.

Microbial cells exist in dynamic equilibrium, coexisting with other cells and their environments[15]. Their metabolic capabilities are encoded in their genomes, but the metabolic programs they execute depend on the differential expression of enzymes[16]. This differential expression enables a variety of metabolic strategies, which are evolvable and can be flexible, heterogeneous, and dynamic[17–19]. For instance, when exposed to a mix of substrates, cells might use these substrates either simultaneously or sequentially[20]—first consuming one and then another—or they might alternate between both strategies[21].

[1]Department of Microbiology, Immunology and Transplantation, Rega Institute for Medical Research, Laboratory of Molecular Bacteriology, KU Leuven, Leuven, Belgium. [2]Université Paris-Saclay, INRAE, PROSE, Antony, France. [3]State Key Laboratory of Regional Environment and Sustainability, Research Center for Eco-Environmental Sciences, Chinese Academy of Sciences, Beijing, People's Republic of China. [4]Key Laboratory of Environmental Biotechnology, Research Center for Eco-Environmental Sciences, Chinese Academy of Sciences, Beijing, People's Republic of China. [5]Unité de Chronobiologie Théorique, Faculté des Sciences, Université Libre de Bruxelles, Bruxelles, Belgium. [6]Department of Chemical Engineering, Chemical and Biochemical Reactor Engineering and Safety (CREaS), KU Leuven, Leuven, Belgium. [7]These authors contributed equally: Daniel Rios Garza, Bin Liu, Charlotte van de Velde. ✉e-mail: danielriosgarza@gmail.com; karoline.faust@kuleuven.be

While the dynamic metabolic strategies of microbes, gene regulation, and phenotype switching have been extensively studied in isolates since Jacques Monod's seminal work[22], their impact on microbiome ecology and stability remains vastly underexplored[23]. In our previous study, we observed that the ecological interactions between two gut bacterial species changed during co-culture[24]. These changes were in response to alterations in pH and the concentration of degradation products resulting from their metabolic strategies. Given that bacteria express alternative metabolic programs under varying environmental conditions, we hypothesize that the coexistence and switching between bacterial growth strategies could induce sharp transitions in community-level phenotypes, leading to multistability and the emergence of predictable alternative community states (Fig. 1). Multistability refers to the capacity of an ecological system to persist in multiple alternative stable states under identical external environmental conditions[25].

In this work, we test this hypothesis both experimentally and computationally by developing and parameterizing a kinetic model. Our experiments confirm the emergence of alternative steady states following transient perturbations in pH and dilution rate. Furthermore, our kinetic model suggests that phenotype switching among subpopulations within the same species is the primary mechanism underlying this behavior.

## Results

### Life history strategies of gut bacteria in a heterogeneous environment

To test this hypothesis, we used a simple three-species gut community that allowed us to combine in vitro experiments with mechanistic modeling. We selected these three species because they represent an important fraction of the adult microbiome, serve as type species for

distinct gut niches, and can reliably grow in our culture medium. We first investigated their individual phenotypes in Wilkins-Chalgren (WC) anaerobic medium. This medium includes two simple carbon sources, glucose and pyruvate, along with substrates from tryptone and yeast extract, which notably contain measurable amounts of trehalose (average 0.71 mM +/−0.07). The composition of this medium is simple enough to allow us to track the kinetics of key metabolites, yet its complex components mimic the nutrient heterogeneity expected in the colon.

We used genome-scale metabolic models to derive sets of biochemical reactions that define the core energy metabolism of each species. We then collected RNA-seq data at different growth stages to confirm the activities of these pathways (Fig. 2A–C). These core pathways connect the import of carbon sources with the production of fermentation acids, enabling us to compare model predictions with the measured data (Fig. 2D–F). By analyzing these pathways alongside live cell growth kinetics, medium pH changes, and metabolite composition, we were able to outline bacterial life history strategies[26]. We incorporated these strategies into an ordinary differential equation model (Supplementary Note 1). This model was calibrated against experimental data, as indicated by the traced lines in Fig. 2D–F. In addition to the pathways outlined in Fig. 2A–C, the full association between genes and metabolic reactions are depicted in Supplementary Fig. 1 and listed in the Supplementary Data 1.

Briefly, under our growth conditions, *Blautia hydrogenotrophica* initially consumes trehalose via a trehalose-specific PTS transporter. The gene for this transporter (*TREpts*) is overexpressed in the early stages of growth compared to the later stages (Fig. 2A). It switches to glucose utilization only after trehalose is depleted, facilitated by a non-PTS glucose transporter (*GLCabc*) that is inhibited during trehalose consumption (Fig. 2A). Interestingly, its genome lacks the glucose-

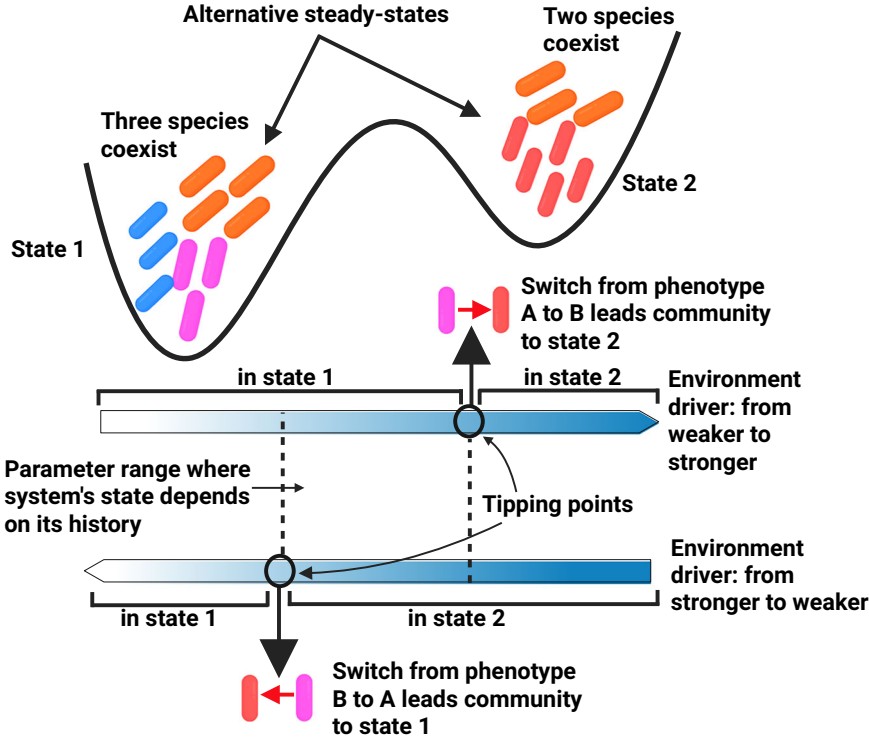

**Fig. 1 | Phenotype switching mechanism that leads to multistability.** According to our hypothesis, in community contexts, multistability can emerge in microbial communities because of phenotype switching between subpopulations of the same species. This switching is triggered by small shifts in steady-state metabolite concentrations, which are influenced by environmental factors such as pH and dilution rate. Because the switching is not necessarily reversible in a symmetrical manner, the system may exhibit history-dependent steady states (i.e., hysteresis), which is a hallmark for multistability. Multistability here means that under the same external conditions the system admits multiple possible stable states (represented as basins of attraction). Created in BioRender. Garza, D. (2025) https://BioRender.com/d46a373.

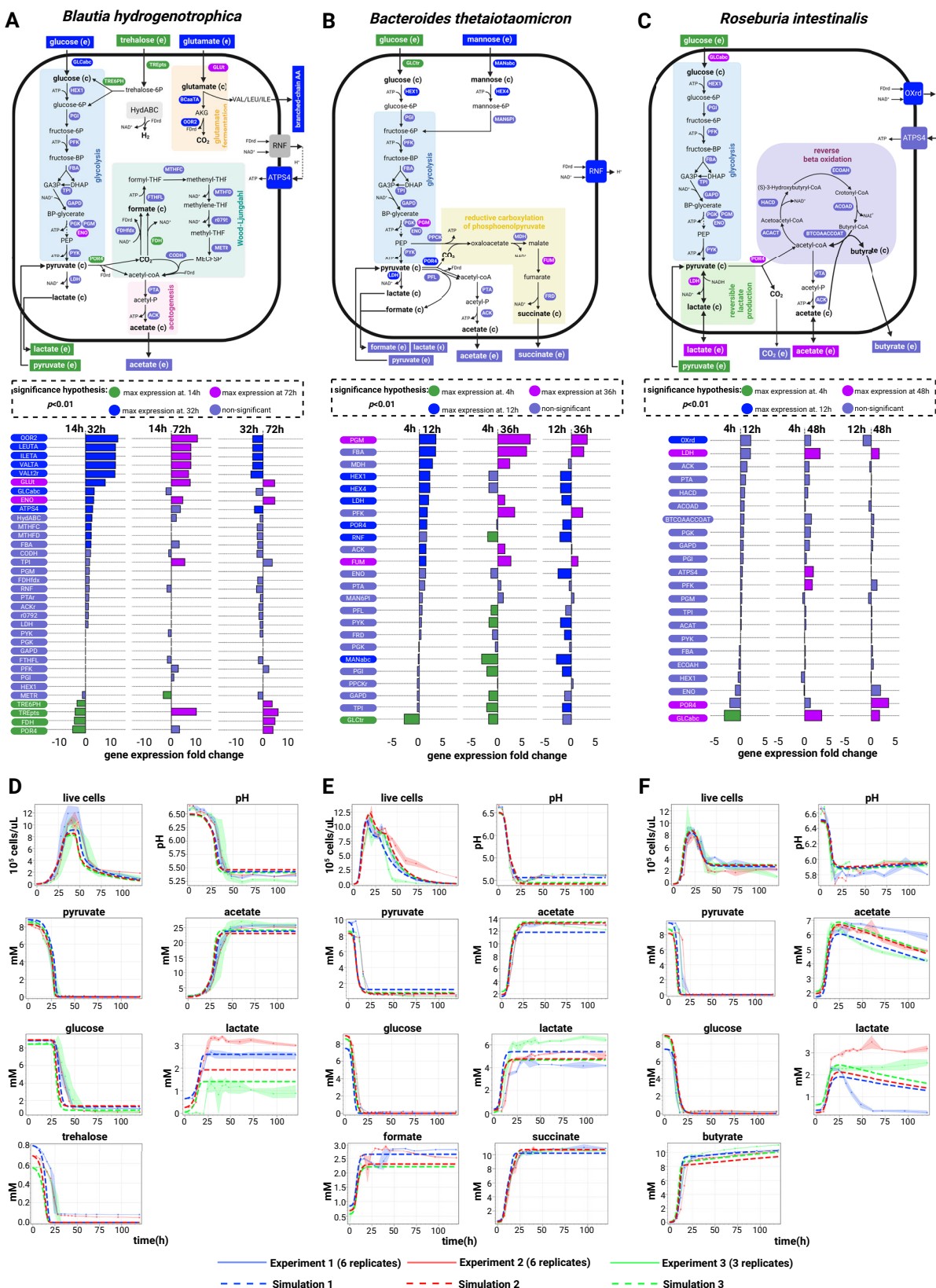

specific IIA component gene of the PTS system, commonly found in closely-related *Blautia* and *Ruminococcus* strains (Supplementary Data 2). We confirmed this sequential substrate preference by showing that higher trehalose concentrations extended *Blautia hydrogenotrophica*'s non-glucose consuming phase (Supplementary Fig. 2). Flow cytometry revealed a distinct bimodal distribution of cell

populations during this metabolic shift, indicative of non-dividing trehalose consumers and dividing glucose consumers (Supplementary Movie 1). Our identification of dividing versus non-dividing cells is based on SYBR-green staining intensity; weaker staining likely corresponds to a non-dividing population and reduced DNA replication activity. The growth rate increased during glucose consumption

**Fig. 2 | Growth kinetics and modeled metabolism of three human gut bacteria.** Schematic illustration of the core energy metabolism pathways for *Blautia hydrogenotrophica* (**A**), *Bacteroides thetaiotaomicron* (**B**), and *Roseburia intestinalis* (**C**), as deduced from genome-scale metabolic model reconstructions coupled with RNA-seq data for temporal pathway activity (see Supplementary Fig. 1 and Supplementary Data 1), and empirical growth data. The middle panel details gene expression changes observed during cultivation, with corresponding gene names linked to their associated reactions in the upper panel. The hypothesis tests were performed using DESeq2. Differential expression was determined using DESeq2 (Wald test, two-sided) with Benjamini–Hochberg correction for multiple comparisons. Exact adjusted *p*-values are reported in Supplementary Data 1. Significance was defined as adjusted $p < 0.01$. The Figure also depicts the consumption of carbon sources (glucose, pyruvate, and trehalose) and the generation of fermentation products (including acetate, lactate, and butyrate). Panels (**D**–**F**) compare experimental growth data over time with model simulations (represented by dashed lines) for the three species cultured in WC medium. The growth data represent averages from three independent monoculture experiments, each with 3–6 biological replicates (solid dots are the averages and the shaded lines indicate the standard deviations of biological replicates). The simulation's initial conditions were the same as the experimental setups and simulations can be reproduced in the Colab Notebook https://github.com/danielriosgarza/hungerGamesModel/blob/main/notebooks/Fig2_simulation.ipynb. Source data are provided as a Source Data file. Created in BioRender. Garza, D. (2025) https://BioRender.com/d46a373.

compared to the trehalose phase (Fig. 2D, ~26 hrs). Equations modeling *Blautia hydrogenotrophica*'s life history strategy are detailed in Supplementary Note 1.

*Bacteroides thetaiotaomicron* rapidly metabolizes glucose and pyruvate, producing fermentation acids that significantly reduce the medium's pH (Fig. 2E). However, this organism is inhibited at low pH conditions (i.e., pH < 5.5)[27]. When the carbon sources are exhausted, most cells lose viability at low pH but can still be detected through flow cytometry. The loss of membrane integrity was confirmed using propidium iodide staining. To reflect this in our model, we introduced functions that describe transitions from active to inactive subpopulations, triggered by nutrient scarcity in acidic environments (see Supplementary Note 1). We consistently observed a second growth peak before a major population inactivation (as shown in Fig. 2E), which we believe is due to trace mannose consumption, consistent with our previous findings[24]. Mannose depletion was verified through measurements and gene expression analysis, although the precise kinetics remain unresolved.

*Roseburia intestinalis* generates butyrate through the reverse β-oxidation pathway (illustrated in Fig. 2C). In our experiments, *R. intestinalis* efficiently consumed glucose and pyruvate, producing butyrate, acetate, and lactate, impacting the pH to a lesser extent than *B. thetaiotaomicron*. As we previously described[24], in the absence of glucose, *R. intestinalis* transitions to a slow growth mode, characterized by the prolonged survival of viable cells (evident in the live cells curves in Fig. 2F), sporadic consumption of lactate/acetate (Fig. 2F), and continuous butyrate production (also in Fig. 2F). In our model, we represented this average behavior by incorporating rapid cell death in the absence of glucose, and by shifting subpopulations to slow growth in response to lactate and acetate (detailed in Supplementary Note 1). However, our model does not fully account for the observed heterogeneous lactate utilization across different experiments (as shown in Fig. 2F).

## The model's stability landscape reveals sharp transition zones

After calibrating our model with monoculture growth data (Fig. 2D–F and Supplementary Note 1) and validating its performance in batch cocultures (Supplementary Fig. 3), we incorporated dilution terms to simulate in silico the stability landscape of the community in a continuous culture environment. We explored how steady-state concentrations of bacteria and metabolites respond to controllable factors—medium pH and dilution rate—which do not directly alter their initial concentrations but can impact the system's dynamics. The dilution rate impacts the steady-state concentration of metabolites and nutrient availability, thereby affecting population growth rates and the transition between metabolic phenotypes (Fig. 3A), while pH directly impacts growth. Changes in growth also impact the production of fermentation acids and, for instance, their subsequent utilization by *Roseburia intestinalis*' slow growth mode. In summary, these parameters significantly shape the overall community phenotype (Fig. 3B–C).

A surprising observation that resulted from this in silico analysis of the community is that the stability landscape of some of the state variables revealed sharp transitions between steady-state metabolite and bacterial cell concentration profiles. For example, *Blautia hydrogenotrophica* maintains consistent concentrations under wide pH ranges, but undergoes a sharp shift when the dilution rate increases beyond a certain level (Fig. 3B). *Roseburia intestinalis* survives in two separate zones, including a narrow range of conditions where it outcompetes the other species (Fig. 3C). These zones are separated by areas where no growth occurs. Overall, the presence of sharp transitions between zones in the stability landscapes suggests that alternative stable states can emerge depending on the initial conditions. It also indicates the presence of tipping points, where minor environmental changes can lead to significant shifts in community structure.

## Phenotype-switching explains multistability in silico and agrees with observations in vitro

To understand the mechanistic basis for alternative stable states in our model, we individually varied the dilution rate while allowing the pH to vary according to the accumulation of organic acids (see Environment pH section of the Supplementary Note 1). Two clearly distinguishable states emerged (Fig. 4A): one where *Blautia hydrogenotrophica* and *Bacteroides thetaiotaomicron* co-dominate (dilution rate <0.041 h⁻¹), with *Roseburia intestinalis* almost completely outcompeted, and another where *Bacteroides thetaiotaomicron* and *Roseburia intestinalis* prevail, with *Blautia hydrogenotrophica* maintaining a low abundance (dilution rate > 0.041 h⁻¹).

In the model, the shift to one state or the other depends on phenotype switching between subpopulations of the same species (Supplementary Fig. 4), specifically, the switch between *Blautia hydrogenotrophica*'s trehalose and glucose-consuming phenotypes. High dilution rates lead to trehalose accumulation, which in turn inhibits the glucose-consuming phenotype. When *Blautia hydrogenotrophica* is not consuming glucose, it occupies a niche that has a negligible impact on the other species. In contrast, if the trehalose concentration is low and a sufficient population shifts to glucose consumption, *Blautia hydrogenotrophica* becomes a strong competitor, inhibiting the other species. Interestingly, this phenotype displays characteristics of hysteresis (ecological memory): once the glucose-consuming population is established, for example, by stopping the feed (setting the dilution rate to zero), then restoring the feed to its previous levels does not return the system to its former state (Fig. 4A). Consequently, two states can coexist under the same parameter values, and the system's history is required to predict its behavior (for a more in-depth exploration of this behavior refer to Supplementary Figs. 4 and 5). We also observed this behavior experimentally using minibioreactors operated as emulated chemostats, with continuous feed consisting of WC medium (Fig. 4C, Fig. 5, and Fig. 6).

Similar alternative stable states also exist along the pH gradient. Because these states depend on the dilution rate, we fixed the dilution

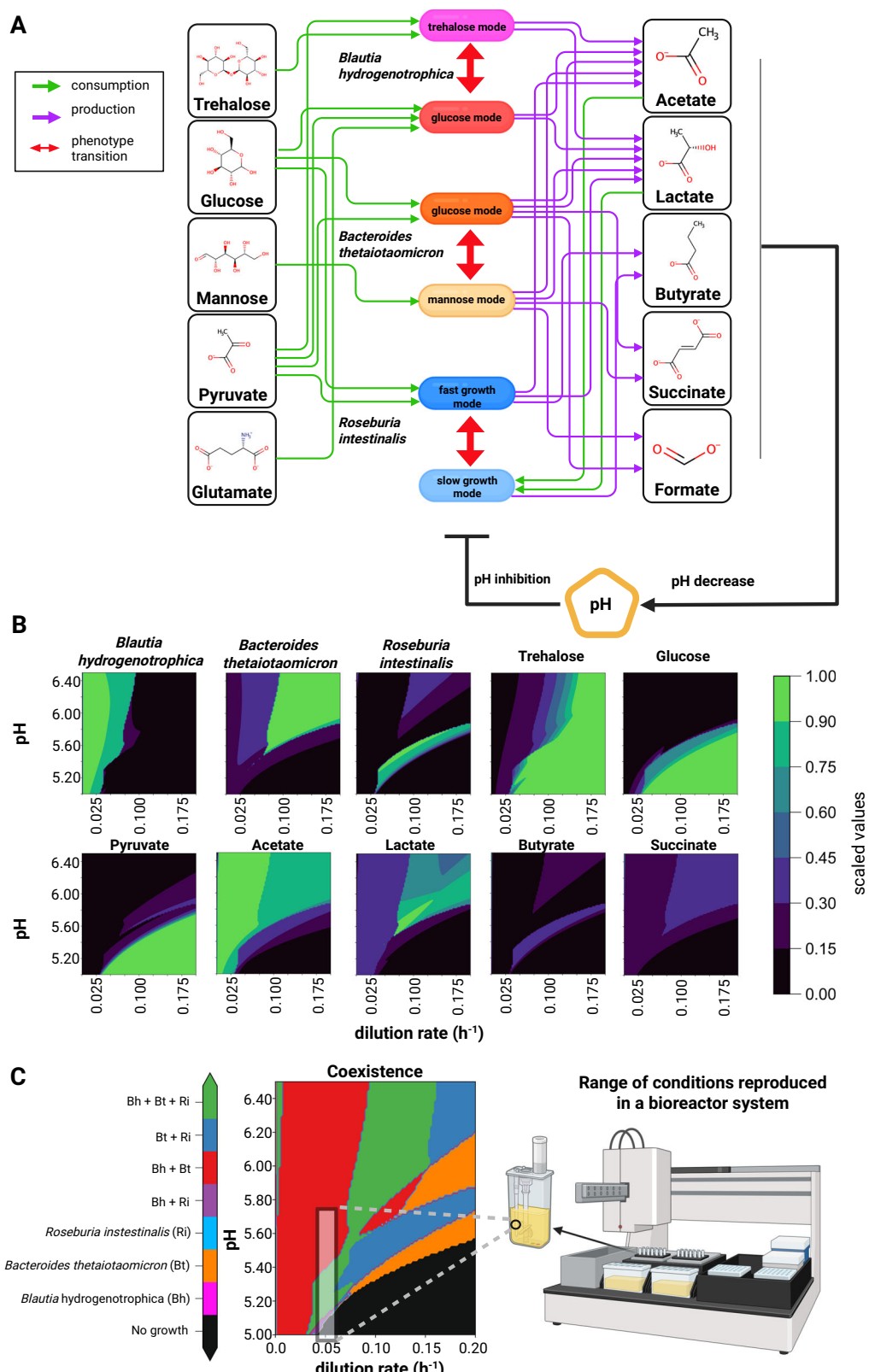

rate at 0.067 h⁻¹ and explored the community landscape with varying pH values (Fig. 4B). At lower pH values (e.g., 5.47 see Fig. 4B), *Roseburia intestinalis* is favored. Overall, *Bacteroides thetaiotaomicron* is favored by pH control, since it can produce a large amount of organic acids without lowering the environmental pH and inhibiting its own growth. The state of the system, however, ultimately depends on its history, which is illustrated in the lower plot of Fig. 4B: changing the pH after

the system is established does not necessarily lead to a change in community state (Fig. 4B). When the transition between subpopulations of the same species are artificially blocked, the system still has different states at different pH ranges since the species vary in their pH sensitivity. However, starting the system at a pH of 5.6 and later changing it to 5.47 leads to the same state as starting it on 5.47 (Supplementary Fig. 4). Notably, this effect is linked to the concentration of

**Fig. 3 | Mechanistic modeling reveals that pH and dilution rate drive community transitions towards alternative stable states. A** Schematic of the mechanistic model encoded as ordinary differential equations (see Supplementary Note 1). The model incorporates experimental data (refer to Fig. 2B–D and Supplementary Figs. 2 and 3) to simulate variable conditions. The arrows indicating consumption and production of metabolites as well as transitions between subpopulations are listed in Table 1. **B** Contour plots depicting the predicted steady-state concentrations of model variables across a pH range of 5.0–6.5 and dilution rates from 0.0 to 0.2 h⁻¹. Concentrations, initially in millimolar (mM) for metabolites and in $10^5$ cells/µL for cells, are normalized to their maximum values for comparative purposes. The colors correspond to a gradient from minimum (black) to maximum values (light green). The maximum values of butyrate and succinate are specifically at dilution rates of 0. **C** A community composition phase plot indicates the presence or absence of species across different pH and dilution rate conditions in model simulations. The black shaded rectangle in the plot highlights the conditions that fall within the range observed in our minibioreactor experiments (refer to Figsures 4, 5, and 6). For all the simulations in panels B and C, we used the starting concentration of 300 cells/µL per species and the average of metabolite concentrations of WC medium that we measured repeatedly at time zero for the experiments shown in Fig. 2D–F (see the Supplementary Note 1). Simulations can be reproduced in the Colab Notebook https://github.com/danielriosgarza/hungerGamesModel/blob/main/notebooks/Fig3_Simulation.ipynb. Created in BioRender. Garza, D. (2025) https://BioRender.com/d46a373.

**Table 1 | Consumption and production of metabolites by subpopulations of *Blautia hydrogenotrophica*, *Bacteroides thetaiotaomicron*, and *Roseburia intestinalis* as encoded in our kinetic model (Fig. 3A)**

|  | B. hydrogenotrophica (trehalose mode) | B. hydrogenotrophica (glucose mode) | B. thetaiotaomicron (mannose mode) | B. thetaiotaomicron (glucose mode) | R. intestinalis (fast growth mode) | R. intestinalis (slow growth mode) |
|---|---|---|---|---|---|---|
| **Trehalose** | Consumes |  |  |  |  |  |
| **Glucose** |  | Consumes |  | Consumes | Consumes |  |
| **Mannose** |  |  | Consumes |  |  |  |
| **Pyruvate** | Consumes | Consumes |  | Consumes | Consumes |  |
| **Glutamate** |  | Consumes |  |  |  |  |
| **Acetate** | Produces | Produces | Produces | Produces | Produces | Consumes |
| **Lactate** | Produces | Produces | Produces | Produces | Produces | Consumes |
| **Butyrate** |  |  |  |  | Produces | Produces |
| **Succinate** |  |  | Produces | Produces |  |  |
| **Formate** |  |  | Produces | Produces |  |  |

cells in a specific phenotype reaching a tipping point and their interactions with metabolites in the medium and not to the extinction of *Roseburia intestinalis* cells, as the shift is performed while there is still a significant abundance of live *Roseburia intestinalis* cells in the system (also see Supplementary Fig. 5).

To test model predictions, we cultured gut species in minibioreactors that enable parallel cultivation and sampling in controlled conditions, emulating a chemostat via continuous inflow and pipette-controlled outflow, as reported previously[28]. When a feed perturbation was applied, the system shifted towards a *Blautia hydrogenotrophica* and *Bacteroides thetaiotaomicron* enriched state as predicted by the model (red triangles in Fig. 5B). These states show sharp transitions in species composition within a narrow range of experimental conditions, as we predicted with the model's stability landscape (Fig. 3C). In reactors where *Blautia hydrogenotrophica* was highly abundant, trehalose was completely consumed (Fig. 4C and Supplementary Data 3), suggesting that the glucose-consuming subpopulation could emerge and influence the abundance of other species. Conversely, *Blautia hydrogenotrophica* was absent or present at low levels in reactors where trehalose remained at detectable concentrations. Of note, these concentrations reflect differences in community-level processing of metabolites in each alternative state, since the inflow of nutrients—the feed—was the same before and after perturbations, and consisted of WC medium.

In summary, perturbations have a lasting effect, even when the system is returned to its previous conditions, including the same continuous inflow of the nutrients in the WC medium. Following only a dilution rate perturbation (feed stop), the system transitioned to a state with high abundance of *Bacteroides thetaiotaomicron* and *Blautia hydrogenotrophica* and low abundance of *Roseburia intestinalis*, similar to the behavior predicted by the model (Figs. 4A and 5B). Following sequential pH and dilution rate perturbations—acidifying the pH to around 4.8 by feeding the reactors with a medium with pH of 3.3 and later stopping the feed (Figs. 4C, 5C, 6 and Supplementary Data 3)—

the replicates diverged into three alternative states after the system was returned to its previous conditions: either reverting to its previous state with low *Blautia hydrogenotrophica* abundance, transitioning to a state were the three species coexist, or moving to a state where *Bacteroides thetaiotaomicron* predominates. This corroborates the history-dependent multistability mechanism suggested by our model's landscape analysis.

## Simulating multistability in systems with many species

To further confirm this multistability mechanism and explore its potential to explain microbiome landscape dynamics, we abstracted key components of our model into a new formalism (for details, refer to Supplementary Note 2). Species are defined by Lotka-Volterra growth rates and interactions, but instead of single growth rates and interactions, species now encompass subpopulations with alternative phenotypes—two and potentially more growth rates and interaction coefficients—with environment-responsive transitions between them (Fig. 7A). As in our mechanistic model, this simplified model exhibits alternative community types separated by a tipping point (Fig. 7B). These states arise from subpopulation shifts to strongly competing phenotypes (e.g., more efficient usage of key nutrients). Simulations show that even in larger communities, environment-driven emergence of such competitive phenotypes can significantly reshape the community landscape, producing distinct community types that resemble enterotypes, i.e., alternative community types observed in fecal samples[3] (Fig. 7C). Similar to an empirical study of species distribution across stool samples[29], the species driving such community shifts—referred to by their original authors as tipping elements—exhibit a bimodal distribution in our model (Fig. 7C).

## Discussion

Here, we explored in depth the metabolic strategies of three prevalent human gut bacterial species and demonstrated in silico and in vitro that multistability is present and according to our model arises as a

**Simulations**

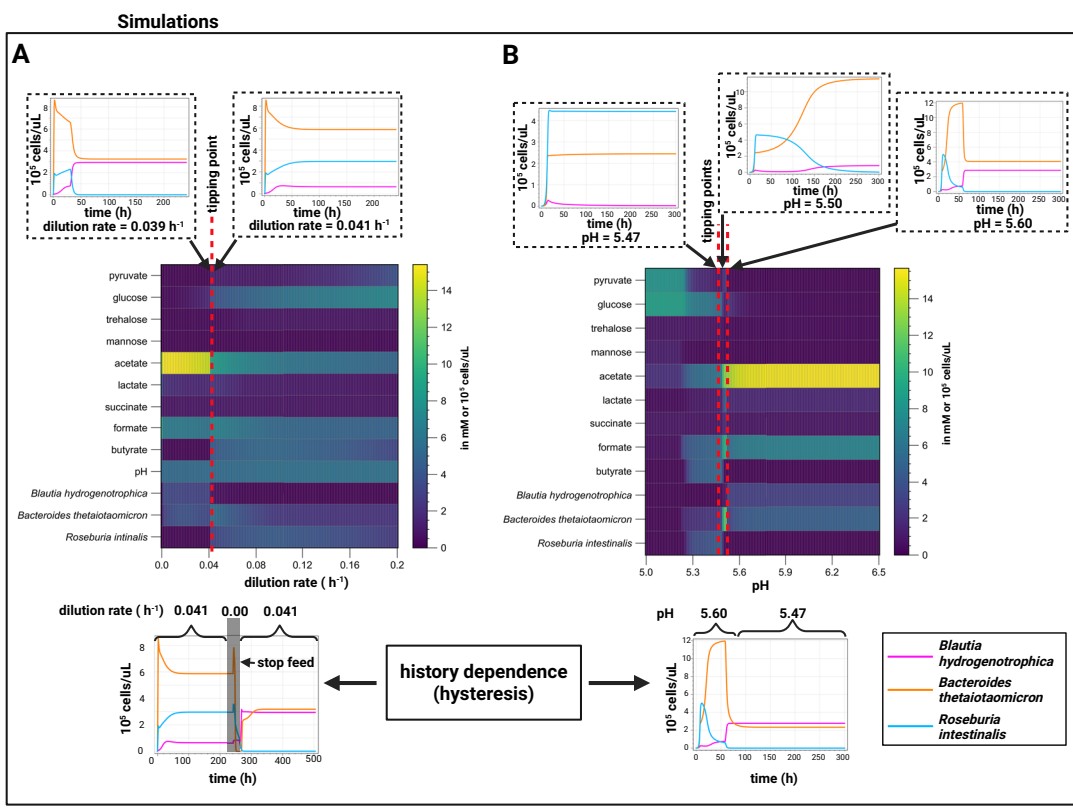

**Observations**

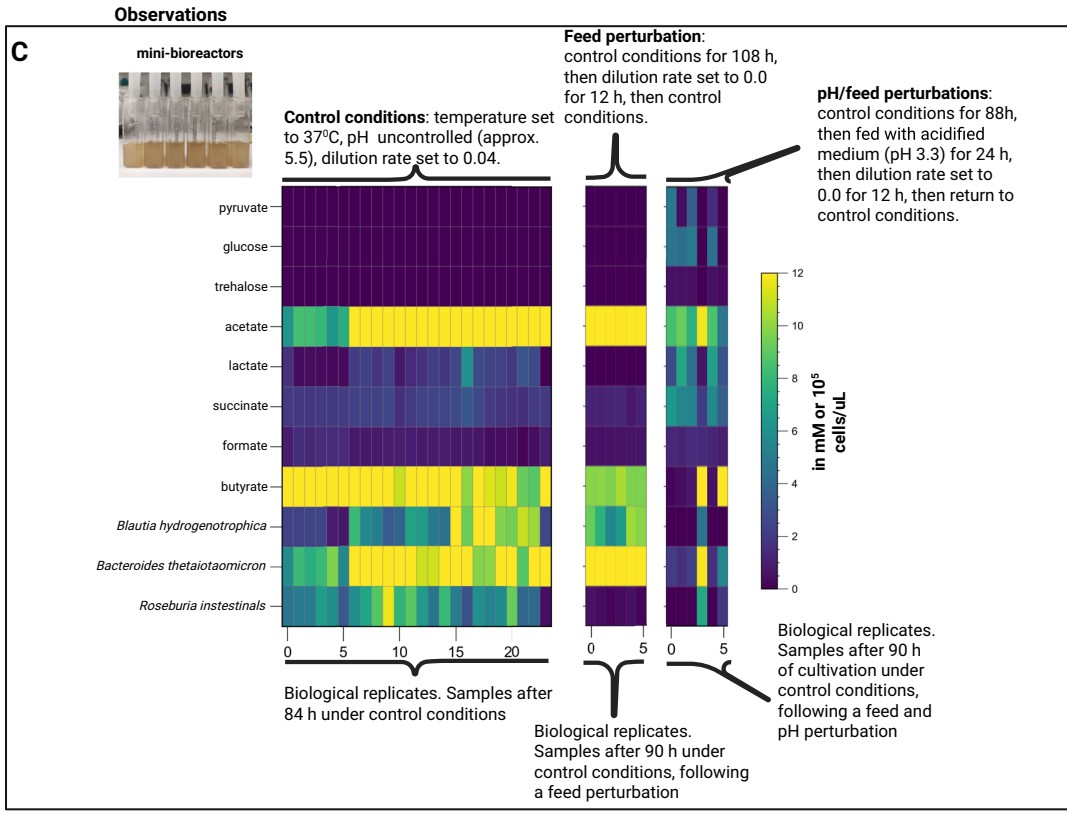

consequence of bacterial metabolic flexibility. Our results suggest that the sharp transitions between alternative states in our system are caused by varying ecological interactions among phenotypically flexible bacteria. For example, we hypothesize that the glucose-consuming phenotype of *Blautia hydrogenotrophica* competes strongly with and inhibits the fast growth mode of *Roseburia intestinalis*. Communities

with a low steady-state trehalose concentration—indicating that this phenotype is active—are significantly different from communities where it is repressed. Of note, we refer to the steady-state concentration of metabolites as the concentration that emerges after the system reaches a steady-state. This concentration results from the system's internal dynamics, rather than its concentration in the feed, which, in

**Fig. 4 | Simulations and experimental observations of tipping points and multistability in a synthetic gut microbial community.** Model simulations predict alternative steady states connected by sharp transitions (tipping points) that exhibit hysteresis (ecological memory). For example, (**A**) changing the dilution rate drives the community toward an alternative state with a sharp transition between 0.039 h⁻¹ and 0.040 h⁻¹. Even when the original dilution rate is restored, the system does not return to its previous state. This scenario, shown in the lower inset, was reproduced experimentally (see feed perturbation in **C**) (**B**) The steady-state community composition that emerges when controlling the system's pH depends on the system's history. When the system starts and remains at pH 5.47, a different state is reached compared to when it starts at pH 5.60 and is shifted to pH of 5.47. When varying the dilution rate (**A**), the pH is an emergent property, whereas in (**B**), it is fixed at specific values. In all the simulations in **A** and **B**, we used the starting concentration of 300 cells/μL per species and average metabolite concentrations from repeated measurements of WC medium. Simulations can be reproduced in the Colab Notebook https://colab.research.google.com/github/danielriosgarza/ hungerGamesModel/blob/main/notebooks/Fig4_simulations.ipynb (**C**) Experimental validation conducted in minibioreactors (shown in the small inset photograph) reveals stable alternative community compositions that persist under nearly identical environmental conditions in an emulated chemostat (see methods), aligning with the model's predictions of multistability. Each column depicts the steady-state reached by a replicate. We systematically collected periodic samples and analyzed community states using HPLC for metabolites, alongside 16S rDNA sequencing and flow cytometry for bacterial species quantification in independent experiments (detailed in Supplementary Data 3, Fig. 5, Fig. 6, and Supplementary Fig. 6). Panel C shows the results from two independent experiments (full time series shown in Supplementary Fig. 6, labeled as A6 and A7 in Supplementary Data 3). In each experiment six vessels among the twelve that were originally cultured under control conditions were successively exposed to feed and pH/feed perturbations (experiments A6 and A7, respectively). Source data are provided as a Source Data file. Created in BioRender. Garza, D. (2025) https://BioRender.com/d46a373.

the chemostat, is held constant. The history-dependent behavior observed in our model and chemostat experiments emerges from feedback loops between subpopulations and environmental factors. For example, while increasing the dilution rate may increase the steady-state concentration of trehalose and inhibit *Blautia hydrogenotrophica*'s glucose-consuming phenotype, if a large population of glucose-consuming cells is already present in the system, then many cells are available to switch and quickly consume the excess trehalose that would otherwise accumulate due to the increase of the dilution rate. This leads to a different proportion of subpopulations and alters the community phenotype. Importantly, these changes have effects similar to those observed when the community is inoculated with vastly different species proportions[30]. However, in our system, alternative states are present in the system's landscape thus shifts between alternative states can also be triggered by changes in environmental parameters, which lead to changes in the associated microbial phenotypes. This mechanism is likely closer to real-world conditions but involves subtle environmental changes that can be difficult to predict.

A limitation of our work is that we did not collect direct evidence of phenotype switching in the community, which requires stable isotope experiments and is a task for the future. Another limitation is that we did not test whether the steady states reached in our chemostat system are robust to invasion. It is possible that a species goes extinct during the transient phase, but that once the system reaches a steady state, the conditions might allow the extinct species to reestablish itself in the community if re-inoculated. This possibility does not arise in the dynamic model, where all three species are always present, even if some persist at exceedingly low concentrations. However, it remains to be tested whether this is also the case in our experimental system.

Several mechanisms beyond environmental variation have been proposed to explain alternative stable states in the gut microbiome. One model by Gibson et al.[6] attributes multistability to priority effects arising from uneven interaction strengths and sub-sampling, but it relies on differing initial community compositions and cannot account for stability shifts following perturbations. Goyal et al.[8] proposed a nutrient-partitioning mechanism based on the stable marriage model, predicting alternative states among *Bacteroides* species depending on nutrient preferences. While conceptually close to our hypothesis, their model is qualitative and does not capture key processes such as metabolite cross-feeding or pH shifts. In contrast, our mechanistic model includes positive feedbacks and non-linearities—conditions required by Thomas' conjecture[31]—and explains how perturbations can induce stable-state transitions.

Our simulations further confirm that the presence of different phenotypes can give rise to alternative community types in large communities. Thus, we suggest that multistability is a potential driver behind alternative community types observed in the gut microbiome[1,3], supporting previous propositions[1,7]. The occurrence of

alternative community types in other host-associated microbiota, which are not easily explained by environmental differences[2,32,33], implies that multistability may be more common than previously thought. As we and others have previously discussed[7,8,29], it is relevant whether alternative community types are due to environmental differences caused by diet, host genetics, or other factors or whether they are due to multistability. In the case of the latter, alternative community types can result from past transient perturbations rather than from current differences between hosts.

We also note that popular mathematical models of microbial communities, such as the generalized Lotka-Volterra (gLV) model, do not account for metabolic flexibility. Moreover, several established methods assume the absence of multistability in microbial communities. One of these is the dissimilarity-overlap curve analysis[34,35], which evaluates the universality of microbial interactions by relying on an empirical negative correlation between compositional dissimilarity and species overlap. Another is EPICS (effective pairwise interactions for predicting community structures), which parameterizes the gLV model from leave-one-out communities[36]. The accuracy of other inference methods such as BEEM[37] or MDSINE[38] may also be affected by the occurrence of multistability.

There are different ways to integrate metabolic knowledge into community models[39]. Here, we opted for a kinetic model instead of a metabolic model. This choice was made to effectively capture pH response and phenotypic switches, as well as to investigate history-dependence and the stability landscape of the community. Our kinetic model was informed by insights from metabolic reconstructions and parameterized using Powell's local optimization method (see Supplementary Note 1). This approach does not ensure unique or globally optimal parameters, and the manually specified parameter ranges may not always reflect biological reality. While we incorporated general mechanisms observed in our experiments, the parameter sweeps (Supplementary Fig. 8) show that some parameters have little or no influence on model dynamics, suggesting that the model can be simplified. It may be possible to construct such kinetic models automatically from metabolic models in the future.

In summary, we have shown that flexible microbial strategies impact the composition of gut microbial communities. In the future, we need to systematically elucidate these strategies in other gut microorganisms[40] to better understand and efficiently modulate gut microbial communities.

## Methods
### Microbial strains
Human gut bacterial strains of *Blautia hydrogenotrophica* S5a33 (DSM 10507 ᵀ), *Bacteroides thetaiotaomicron* VPI-5482 (DSM 2079 ᵀ) and *Roseburia intestinalis* L1-82 (DSM 14610 ᵀ) were obtained from the Deutsche Sammlung von Mikroorganismen und Zellkulturen (DSMZ,

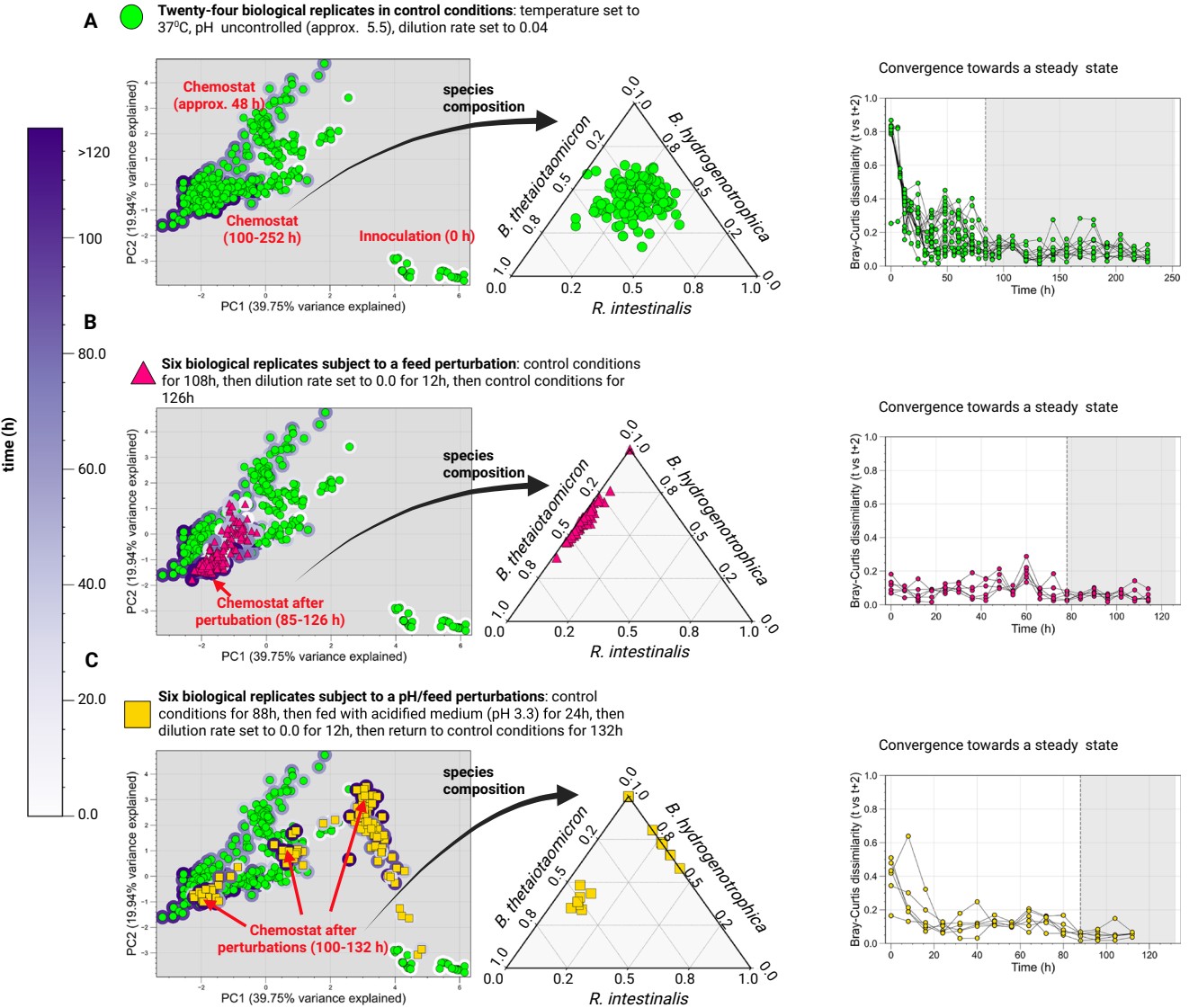

**Fig. 5 | Minibioreactor experiments reveal multistability following feed and pH perturbations.** PCA plots were generated from metabolite and cell concentration data collected across all time points and biological replicates from two independent experiments (24 biological replicates for the control group, 6 for the feed perturbation, and 6 for the pH/feed perturbation; in the control group, we counted the twelve samples that were later exposed to perturbations. Feed and pH/feed perturbations were carried out on times 108 and 88, respectively). Steady-state points were identified based on the Bray–Curtis dissimilarity between each time point and the corresponding metabolite and cell concentrations two steps ahead (t and t + 2), selected ranges are highlighted in gray. To ensure comparability across variables with different units and dynamic ranges, each variable (metabolite or cell type) was normalized using min–max scaling to a [0, 1] range. The time series for these experiments are shown in Supplementary Fig. 6. Minibioreactors were operated as emulated chemostats, containing WC medium as the feed. **A** Control samples converge to a steady state in which *Blautia hydrogenotrophica*, *Bacteroides thetaiotaomicron*, and *Roseburia intestinalis* coexist, as shown in the ternary plot.

**B** A feed perturbation drives the system to a state in which *Roseburia intestinalis* is outcompeted, consistent with predictions from the mechanistic model (Fig. 4A). **C** A pH perturbation followed by a feed perturbation leads to three distinct alternative states inferred from the different community compositions visible on the ternary plots, which show consistently low Bray–Curtis dissimilarities between consecutive time points (t and t+2). The bioreactor experiments shown here were initially inoculated with five species. Absolute abundance data showed that the fourth species *Faecalibacterium duncaniae* remained at low levels and that the fifth species *Segatella copri* did not establish (see Supplementary Fig. 6). Although bacterial quantification was based on absolute abundance, we cannot rule out that the additional species influenced community dynamics. To confirm the multistable outcomes, we repeated experiment (**C**) using only *Blautia hydrogenotrophica*, *Bacteroides thetaiotaomicron*, and *Roseburia intestinalis*, and recovered two of the three steady states previously observed (Fig. 6). Source data are provided as a Source Data file. Created in BioRender. Garza, D. (2025) https://BioRender.com/d46a373.

Germany). The strains were frozen in Wilkins-Chalgren Anaerobe Broth (WC; Oxoid Ltd., Basingstoke, United Kingdom) plus 20% glycerol and maintained at -80 °C until use.

### Batch cultivation and sample collection
Batch cultivations of monoculture, bi-culture and tri-culture were followed for 120 h in 120-ml serum bottles containing 60 ml of WC medium. The serum bottles were prepared in the same way as

previously described[24], and were inoculated with 1 ml of the diluted preculture to an OD600 of 0.1 (either a single species or the mixture of them). The bottles were incubated at 37 °C and at a constant stirring rate of 170 rpm (shaker KS 4000 i; IKA, Staufen, Germany). Samples were taken from the liquid broth every four hours for the first 48 h and every 12 h afterwards. Three biological replicates were designed for testing the monocultures in three independent batch experiments. All bi-culture and tri-culture experiments were performed in six biological

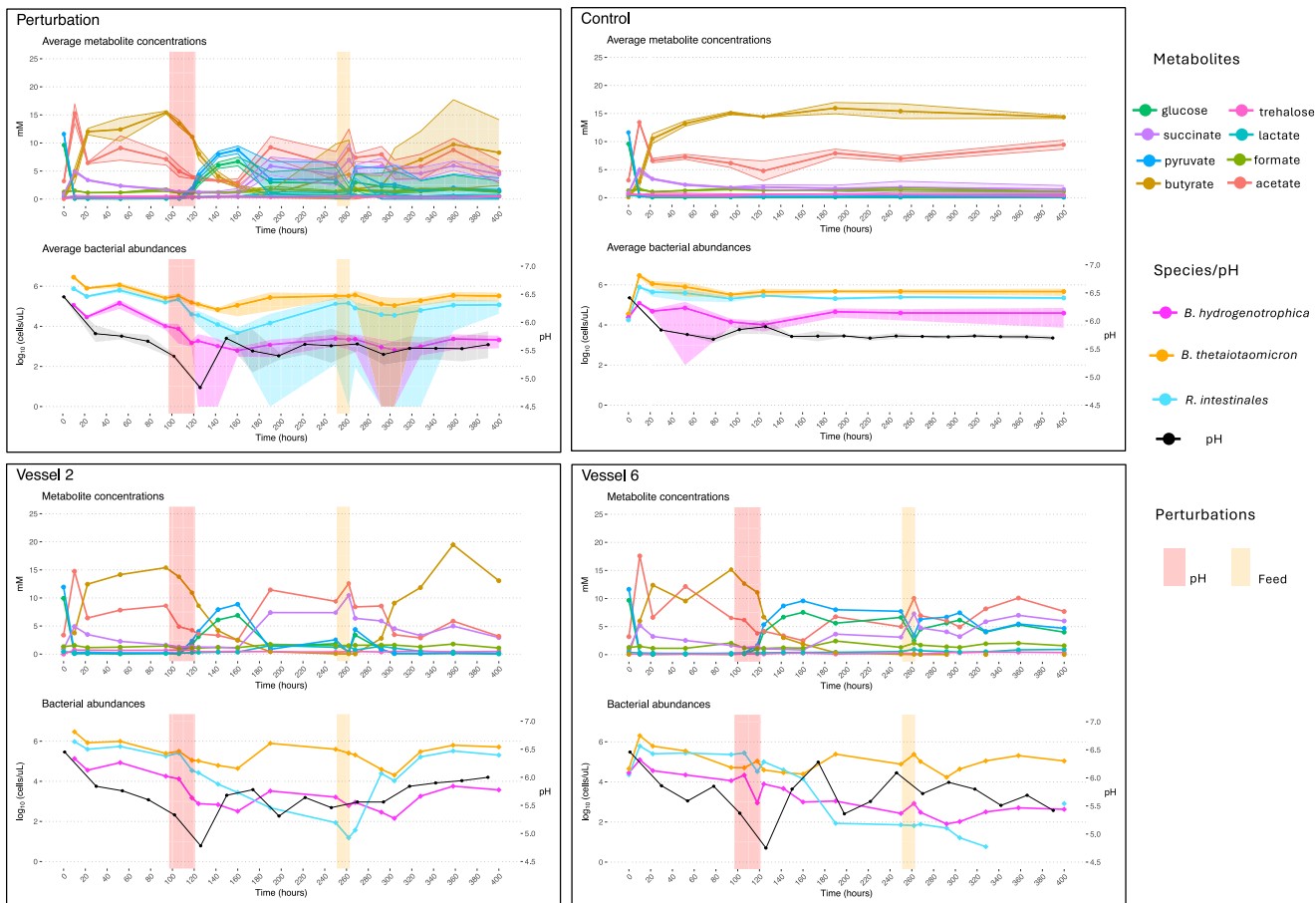

**Fig. 6 | Metabolite concentrations and bacterial abundances over time for replicate experiments in minibioreactors.** The control consists of six biological replicates (three for metabolite concentrations), while the perturbation contains eight replicates (three for metabolite concentrations). Data represented as mean values +/− SD. In the first four hours, minibioreactors were operated in batch mode, then were switched to chemostat mode with a continuous feed of WC medium. For illustrative purposes, we highlight the time series of two replicates that converged to alternative states. Of note, perturbed and control minibioreactors were all inoculated from the same source and monitored in parallel. The full dataset is available in Supplementary Data 3 and time series plots for additional mini-bioreactor experiments are shown in Supplementary Fig. 6, we also show PCA trajectories and ternary plots in Supplementary Fig. 7. Source data are provided as a Source Data file.

replicates and always had a negative control bottle without inoculation but with sampling for each time, to verify its sterility.

Sterile syringes were used for each timepoint to collect 1 ml of the fermentation broth into 2-ml tubes (Eppendorf) under anoxic conditions. Subsequently, these tubes were used to measure OD600, pH, metabolites and to count bacterial cells by live/dead staining followed by flow cytometry.

**Chemostat experiment with ambr 15**

Fermentations were performed in the Ambr® 15 Fermentation system (Sartorius Stedim Biotech, Royston, UK) located inside a Don Whitley A155 Anaerobic Workstation with HEPA filter (10% $H_2$, 10% $CO_2$, 80% $N_2$, 55% humidity) as previously described[27]. Strains reported were pre-cultured first for 48 h, in modified Gifu Anaerobic Medium broth (mGAM, HyServe) then cultured for 18 h in WC (1/100th dilution without washing) before inoculating the minibioreactors.

Prior to inoculation, strains were diluted in WC medium, mixed in even ratios based on $OD_{600}$ and inoculated in the minibioreactors to a total volume of 10 ml (OD 0.001 for *Bacteroides thetaiotaomicron* and 0.002 for the others). A sample was taken at time point 0 and continuous feeding and sampling started at 4 h after inoculation. The feed consisted of WC anaerobe broth and was delivered at an approximate rate of 0.04 $h^{-1}$, resulting in a complete change of the medium in 24 hours. Samples (250 µl) were pipetted into a cooled plate (4 °C).

In the first two of the three experiments (Supplementary Data 3) *Segatella copri* (DSM 18205, previously Prevotella) and *Faecalibacterium duncaniae* (DSM 17677, previously *F. prausnitzii*) were also inoculated into the system. However, they maintained negligible abundance. We report absolute abundances based on flow cytometry measurements and confirmed the main observation of multistability following a sequential pH and feed perturbation inoculating the system with only *Blautia hydrogenotrophica*, *Bacteroides thetaiotaomicron*, and *Roseburia intestinalis*.

In the first experiment, we first applied a pH perturbation by decreasing the pH of the feed (WC with pH 6.4 to pH 3.7) between 88 and 112 h (24 h total). Subsequently, we applied a perturbation in the dilution rate by stopping the feed for 12 h between 150 and 162 h (periodic removal of liquid continued), after which an additional 5 mL of fresh medium (50% of total volume) was added to the vessels.

In the second experiment, we applied only a feed perturbation between 108 and 120 h (12 h) as described above, including the 5 mL addition of fresh medium (Supplementary Fig. 6).

In the third experiment, we applied a pH perturbation by decreasing the pH of the feed (WC with pH 6.4 to pH 3.38) between 97 and 109 h (24 h total). Subsequently, a feed perturbation was applied between 251 h and 263 h (so during 12 h) as described above including the addition of fresh medium.

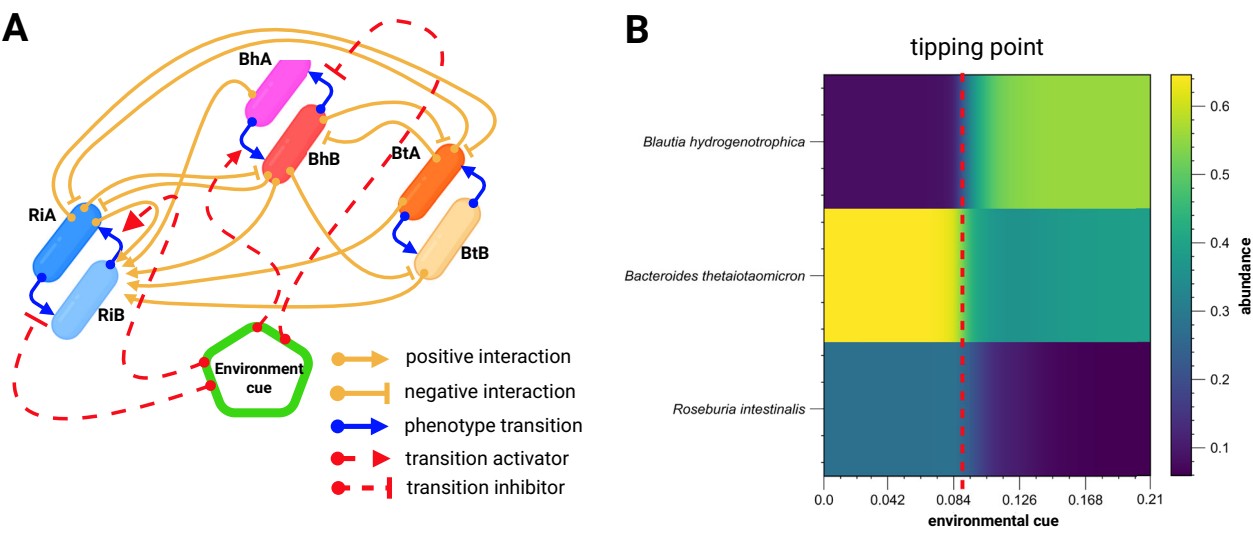

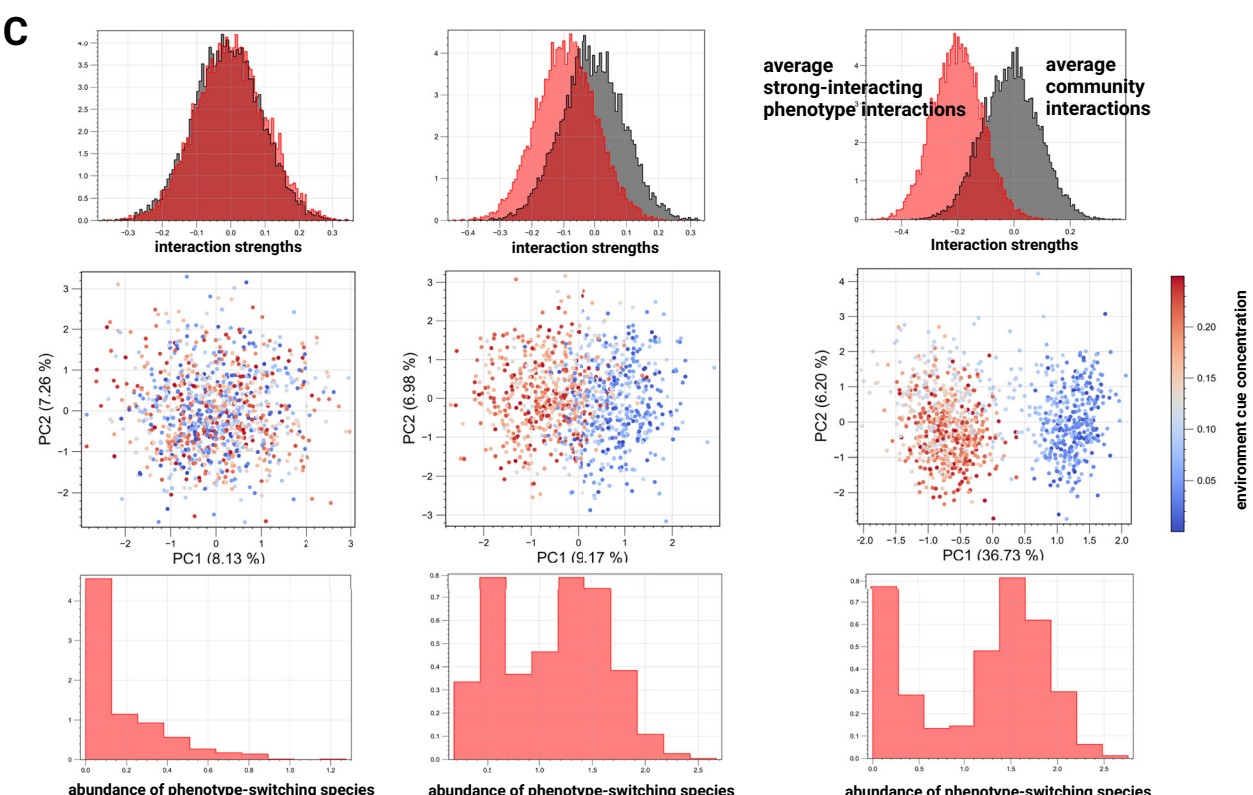

The 16S rRNA genes of selected samples was extracted and sequenced as previously described[28]. Additionally, we used Nanopore MinION flow cell sequencing in our last minibioreactor experiment[41] (Fig. 6). For this purpose, genomic DNA was extracted using Zymo-Biomics DNA miniprep kit (Zymo, USA) with slight modifications to prevent biased extraction. After DNA extraction, the full-length 16S rRNA genes were amplified through a single PCR program. Zymo Biomics Microbial Community DNA standard (Zymo, USA) is included in one well per sequencing plate as the quality control to avoid any bias in different sequence libraries. The other operations were carried out according to the protocol of EasySeq™ Full-length 16S Library Prep Kit (NimaGen, Netherlands). The sequence library was then loaded onto

**Fig. 7 | Toy model demonstrates conceptual mechanism for multistability.**
**A** Species interact through alternative phenotypes connected by environment-responsive transition functions, implemented through Hill equations, allowing dynamic switching between phenotypes during simulations (refer to Supplementary Note 2 for details). **B** If one phenotype strongly interacts with others (average interaction strengths are higher than the average community interactions), phenotype switching can induce a sharp transition between alternative community states (e.g., high steady-state trehalose leads *Blautia hydrogenotrophica* to a weakly competing phenotype, but low trehalose concentration triggers a metabolic shift, enabling *Blautia hydrogenotrophica* to strongly outcompete others in glucose).
**C** Simulations with 1000 random communities containing 50 species and random

concentrations of an environmental factor show that this mechanism can explain emergent alternative stable states reminiscent of enterotypes (visualized as two distinct clusters in principal component space). Gray histograms show the distribution of interaction strengths across communities; red histograms show the distribution of interactions of a strongly interacting phenotype that is expressed in response to the concentration of an environmental factor. The lower histograms show the distribution of abundance of the species expressing the strongly interacting phenotype across samples. Notably, when enterotypes emerge, the switching species tends to exhibit a bimodal distribution across samples. Created in BioRender. Garza, D. (2025) https://BioRender.com/d46a373.

the Nanopore MinION flow cell and sequenced using MinKNOW (version 24.02.8). The minimal read-length was set to 1000 bp and real-time basecalling was performed with the super-high accuracy model to improve sequence quality. The barcoding score was set to at least 70 to demultiplex and trim the barcode afterward. The long-read classifier minimap2 was employed with the NCBI RefSeq 16S rRNA database for taxonomic classification[42,43]. The 16S rRNA gene copy number database rrnDB was used to correct the different 16 s rRNA copy numbers of the strains in the community[44].

### Quantification of live cells with flow cytometry
We used a combination of the DNA-based stains SYBR Green I (SG; Invitrogen) and propidium iodide (PI; Invitrogen) to stain bacterial cells with intact and damaged cytoplasmic membranes[45]. Under anoxic conditions, cells were diluted in filter-sterilized PBS buffer. 1:10 for the first two time points (0 and 4 h) and 1:200 for the next points and stained with a saturating solution of SG/PI, incubated for 20 min in the dark at 37 °C right and immediately measured by flow cytometry using the benchtop CytoFLEX S flow cytometer (Beckman Coulter, Brea, USA) instrument. Events were recorded for exactly 1 min at a sample flow rate of 10 μl/min, applying threshold values of 3000 and 2000 for the forward and side scatters, respectively, values that we have previously validated[24]. We also used 0.5 μm and 1 μm green fluorescent beads (Thermo Fisher Scientific, USA) as internal standards.

### Flow cytometry data analysis
We used an in-house developed pipeline to accurately quantify the absolute abundance of live *Blautia hydrogenotrophica*, *Bacteroides thetaiotaomicron*, and *Roseburia intestinalis* cells for batch cultures. We clustered flow cytometry events in a UMAP space, following a detailed protocol available at: bit.ly/3WNrslL. Raw flow cytometry data were normalized, scaled, and transformed using the arcsin function. The data was then projected into three-dimensional UMAP space, allowing us to classify cell populations into four categories: live, inviable, debris, and blank. This classification was based on empirically determined thresholds for propidium iodide (PI) and SYBR green (SG) signals and by distinguishing from blank control runs.

Further analysis involved using monoculture samples for species classification in co-culture samples. We created a training space with labeled UMAP projections of random events from each monoculture replicate. Supervised UMAP and the *K*-nearest neighbors vote classifier (parameters: n_neighbors = 50, weights = distance, and metric = Mahalanobis) were employed to assign species labels to co-culture events. Prior to finalizing classifications, each sample was overlaid with corresponding blank runs for manual verification using clickable three-dimensional scatter plots, ensuring accurate separation of cell populations from blanks. This process also validated our parameter choices for UMAP and the classifier. Additional details and scatter plot examples are available at: bit.ly/3WNrslL.

### Metabolite profiling
After centrifuging the liquid broth for 20 min at 21,130 × *g* at 4 °C (Centrifuge 5424 R; Eppendorf, Hamburg, Germany), the supernatants

were used for measuring the concentrations of trehalose, glucose, pyruvate, succinate, formate, acetate, lactate and *n*-butyrate, which were determined by high-performance liquid chromatography (HPLC) as previously reported[24]. We also measured the first and end time points of the blank controls. Metabolites propionate, *iso*-butyrate, *n*-valerate and *iso*-valerate were measured but their concentrations were not consistently different from the blank WC control. Most of these metabolites are found in our system as organic acids. In the text, however, for brevity we refer to them by their salt names.

### RNA extraction and sequencing
A total of 27 samples representing the different growth phases of monocultures in three biological replicates were selected for RNA sequencing, including timepoints of three independent experiments: monoculture *Blautia hydrogenotrophica* in WC (14 h, 32 h and 72 h), monoculture *Bacteroides thetaiotaomicron* in WC (4 h, 12 h and 36 h) and *Roseburia intestinalis* in WC (4 h, 12 h and 48 h). Details of the extraction and purification of total RNA, evaluation of RNA integrity and yield, as well as library preparation and sequencing can be found in our recent paper[24]. Although we used the same methodology, the RNA sequencing data of *Blautia hydrogenotrophica* is unique to the current study.

### RNA-seq data processing
Initially, low-quality reads and adapters were removed using fastp. We then mapped high-quality RNA reads to reference transcripts using Salmon in selective alignment mode, employing a decoy-aware index constructed from each organism's genome. We used the latest reference genomes and transcripts from the BV-BRC database. Starting from the counts files, we used the R package DESeq2 (https://bioconductor.org/packages/release/bioc/html/DESeq2.html) to estimate the *p*-values between pairs of conditions. The R scripts for these analyses can be found here: https://github.com/danielriosgarza/hungerGamesModel/blob/main/scripts/R/.

To integrate the gene expression data with genome annotations and metabolic information from the genome-scale metabolic models, we wrote a Python class, which is available here: https://github.com/danielriosgarza/hungerGamesModel/blob/main/scripts/geneExpression/parseGenExpData.py. With this class one can, for example, draw the bar charts of Fig. 2A–C.

### Modeling
In this manuscript we built two computational models based on ordinary differential equations: a mechanistic model based on metabolite and cells kinetic equations (depicted in Fig. 3A) and a phenomenological model based on the generalized Lotka-Volterra dynamics (depicted in Fig. 7A). A detailed description of the model parameters, rationale, and experimental validation is available in the Supplementary Note 1 and 2. A detailed implementation of the reported simulation and code to reproduce all of our computation analysis is available at the project's Github repository[46]: https://github.com/danielriosgarza/hungerGamesModel, which also contains a comprehensive Wiki to help users reproduce our analysis, and detailed

instructions to reproduce all the manuscript Figures in a Jupyter notebook (https://github.com/danielriosgarza/hungerGamesModel/tree/main/notebooks).

## Reporting summary

Further information on research design is available in the Nature Portfolio Reporting Summary linked to this article.

## Data availability

All data the data and code used in the analysis is available at the GitHub repository[46]:   https://github.com/danielriosgarza/hungerGamesModel, which include detailed instructions to reproduce all the Figures in the manuscripts (https://github.com/danielriosgarza/hungerGamesModel/blob/main/notebooks/allFigsManuscript.ipynb). The raw RNA sequencing data were deposited on NCBI's Sequence Read Archive with the accession numbers: SAMN32321133-38 [https://www.ncbi.nlm.nih.gov/bioproject/PRJNA914119/]. SAMN39333017-19 [https://www.ncbi.nlm.nih.gov/bioproject/PRJNA1063153/]. The raw 16S rRNA gene sequencing data from the minibioreactor experiments were deposited on NCBI's Sequence Read Archive with the accession numbers: PRJNA1189023; PRJNA1197443; PRJNA1197391. Raw flow cytometry data were deposited on the Flow Repository with accession numbers: FR-FCM-Z6YM; FR-FCM-Z6YN; FR-FCM-Z74P; FR-FCM-Z753; FR-FCM-Z754. Source data are provided with this paper.

## Code availability

All the data and code used in the analysis is available at the GitHub repository[46]: https://github.com/danielriosgarza/hungerGamesModel, which includes detailed instructions to reproduce all the Figures in the manuscript (https://github.com/danielriosgarza/hungerGamesModel/blob/main/notebooks/allFigsManuscript.ipynb).

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

## Acknowledgements

We thank Gertjan Gerits for helping in the experiments shown in Fig. S1B and Aristeidis Litos for some useful discussions regarding the model in Fig. 4A. We would like to thank Emma Hernandez-Sanabria, Veronica Lloréns-Rico and Jelle Matthijnssens for input and support concerning bacterial transcriptomics, as well as Jeroen Raes for access to the CytoFLEX. We are also grateful to Thi Thuy Duyen Nguyen, Leen Rymenans and Geert Huys for assistance in flow cytometry, 16S rRNA gene library preparation and anaerobic cultivation, as well as to Raul Yhossef Tito Tadeo for processing raw sequence reads. This work was supported by funding from the Research Foundation—Flanders (grant no. G0I0918N and no. G046721N, K.F.) and from the European Research Council (ERC) under the European Union's Horizon 2020 research and innovation program under grant agreement no. 801747, K.F.

## Author contributions

Conceptualization: K.F., D.R.G., C.V., D.G.; Data curation: B.L., C.V., D.R.G., K.S.; Formal analysis: D.R.G., B.L., K.S., C.V. Funding acquisition: K.F., K.B. Investigation: B.L., C.V., D.R.G., P.S., X.Z. Monoculture experiments: D.R.G., B.L. AMBR Experiments: C.V., P.S., X.Z. Metabolite measurements: K.S.; Methodology: all authors. Project Administration: K.F. Resources: K.F., K.B., K.S. Software: D.R.G., D.G. Supervision: K.F. Validation: all authors. Visualization: D.R.G., B.L. Writing—original draft: D.R.G., K.F., B.L. Writing—review & editing: all authors

## Competing interests

The authors declare no competing interests.
