## [Transparent Peer Review file · Nature Communications]

Emergence of alternative states in a synthetic human gut microbial community

Corresponding Author: Professor Karoline Faust

Version 0:

Reviewer comments:

Reviewer #1

(Remarks to the Author)

This manuscript represents a novel and highly significant contribution to the field of microbiome research. The study uniquely integrates metabolomics and species abundance data to investigate alternative stable states in a synthetic human gut microbiome. The experimental validation of multistability alongside the mechanistic modeling provides insights into community-level dynamics.

This work is the first to study alternative stable states in human gut microbiomes with this level of mechanistic detail, measuring both metabolite concentrations and species abundances as a function of chemostat dilution rate and pH of the medium.

While the study is scientifically robust, the clarity of the results presented in Figures 2- 4 is a critical issue. These figures are central to the paper's main message, yet their current presentation makes the results difficult to understand. The modifications recommended below are presented in order of their relative importance.

Recommendations for Revisions:

1. The experiment presented in Figure 3C is completely unclear. Provide a detailed description of what happened over time and at which specific time points. Clearly define what constitutes the control conditions.
2. The central message of the paper—transitions between alternative stable states—is poorly represented in Figures 3D and 3E. It is essential to show these transitions more clearly. Where are alternative stable states in Figure 3E? Where can I see memory effects in transitions between these states in the experimental part of Figure 3?
3. Using greyscale shadows of different intensity in Figures 3D and 3E is ineffective. Incorporating a gradient in intensity of colors will better depict changes in states over time.
4. The use of bars to denote species abundances in three-species communities in Figure 4 is ineffective. These bars are too small to discern differences in species composition. I recommend switching to a log-scale for the Y-axis to better represent species abundances and use line+symbols plots similar to those used for control panels.
5. Figure 2A, which illustrates phenotype transitions and metabolic interactions, is difficult to interpret due to the "spaghetti-like" lines connecting species and metabolites. It is nearly impossible to trace how different phenotypes transition and how these transitions modify which metabolites are consumed or produced by each of three species. A more effective alternative might be to present this information as a table listing each phenotypic state along with the metabolites each species consumes and produces. This would provide a clearer view of the key interactions driving the alternative stable states.
6. The paper reports phenotypic transitions in metabolic utilization and production of metabolic byproducts. These include but are not limited to diauxic shifts, e.g. the switch from trehalose to glucose utilization by *Blautia hydrogenotrophica*. Diauxic shifts were responsible for alternative stable states in a stylized model described in Ref. 8. Highlighting the connection and differences between the results of this study and existing models of alternative stable states in human gut microbiome in the

discussion would strengthen the theoretical foundation of the study and contextualize these findings within broader ecological frameworks.

7. Panels in Figure 4 lack labels, making interpretation challenging. Add clear labels to each panel and use distinct colors for metabolites and species to prevent confusion.

8. Figure 3C is missing an X-axis label, and Figure 2C is missing a Y-axis label. Ensure that all figures have properly labeled axes to improve clarity.

Conclusion: This manuscript is a groundbreaking study that advances the understanding of microbial community stability and dynamics. The findings are highly significant, but the presentation of key results requires substantial improvement. Addressing these issues will greatly enhance the accessibility and impact of the paper. I strongly recommend acceptance after these revisions.

(Remarks on code availability)

Reviewer #3

(Remarks to the Author)

Garza et al., the authors demonstrated that phenotype switching between subpopulations within the same species is responsible for multi-stability in bacterial communities. They first explored the life history strategies of three gut bacterial species using RNA sequencing and metabolomics in monoculture. Next, they developed a parameterized kinetic model to predict community structure. Finally, they utilized a simple three-species consortium in a mini-bioreactor to validate their findings on phenotype switching and alternative stable states.

Although the study is quite informative, appealing, and worthy of publication in Nature Communication, the authors need to clarify certain sections of the article. My comments are detailed below, organized by section of the paper.

Abstract & Introduction:

The author's should rephrase "Thus, a transient perturbation combined with metabolic flexibility is sufficient for alternative communities to emerge, implying that they are not necessarily explained by differences between individuals." As the sentence itself implies, the difference in individuals has nothing to do with community composition, which is not demonstrated in the paper. I recommend rephrasing this sentence.

Most of the introduction follows a chronological order; I would still like to see a summary paragraph at the end. The introduction concludes by stating the hypothesis, which seems somewhat unusual.

Result 1 (Life history strategies of gut bacteria in a heterogeneous environment):

- Is there a rationale for selecting these three specific bacterial species? Was the selection random, or was it based on the bacteria's ability to consume distinct nutrients? A brief explanation would be helpful.
- For Fig. 1A, why are there two panels labeled 14h/32h? Did you mean 14h/72h?
- The figure legends should be more informative. They need to specify the number of replicates (for example, by including numbers in brackets) and clearly explain the significance of the value cutoffs and any statistical corrections used. This suggestion applies to all figures in the manuscript.
- It would also be useful if the authors provided a summary of the RNA-seq data. Are there additional differentially expressed genes? What percentage of these genes are related to nutrient utilization beyond the predicted targets? While the focus is on the expected subset of genes, the current text implies that only those genes are significant. A supplementary panel summarizing these data would improve clarity.
- How confident are the authors that no other nutrient utilization occurs in monoculture? Although untargeted metabolomics does not provide complete annotation, it can still demonstrate that the predicted sugars drive growth.
- In Supplementary Movie S1, the authors claim bimodal growth. While I agree that two distinct subpopulations are present, how do the authors justify the statement regarding "the presence of similar subpopulation sizes of non-dividing trehalose consumers and dividing glucose consumers"? It appears that after the initial bimodal separation, the subpopulations continue to divide, as indicated by an increase in density. Plotting the number of cells in the two separate clusters after the separation would provide clearer evidence.

Result 2 (The model's stability landscape reveals sharp transition zones):

- Figure 2A is unclear because the central cylinders are shown in multiple colors. Do these colors indicate specific attributes? It appears from legend that two of the cylinders correspond to one bacterium. Consider making this explicit by labeling the bacteria with corresponding colors on the cylinders. The figure legend also needs to be more detailed and clearer.
- For Figure 2B, please include a labeled legend for the gradient colors, similar to the other figures.
- Overall, the results are well presented up to Figure 2B. However, the sudden introduction of the bioreactor systems later in the results disrupts the narrative flow. It is recommended to either move the bioreactor figure to a later section or provide a clearer explanation of the experiment.

Result 3 (Phenotype-switching explains multistability in silico and agrees with observations in vitro)

- For Figure 3, the lower inset requires clearer annotation; the authors should label this inset as Fig. 3C. The manuscript states that hysteresis influences community composition, but it is unclear whether this effect is driven by the initial inoculum concentration or by the metabolic memory of the cells. A bioreactor experiment using an inoculum that mimics post-perturbation conditions could help clarify this issue and might also explain the three distinct steady states observed during

pH perturbation. Additionally, highlighting or circling the two replicate clusters in the PCA would enhance clarity.

- Regarding Figures 4A and 4B, while these are mentioned in the text, the figures themselves lack adequate annotation. The authors should consider labeling the panels sequentially (e.g., from Figure 4A to 4L) and using these labels consistently throughout the results section.
- The rationale for including a 4-species community should be more explicitly detailed in the results section. The authors need to explain why this community was chosen and what motivated the experiment.
- Figure S5 is not referenced anywhere in the results. The authors should cite this figure in the text.

(Remarks on code availability)

The Github page is well-organized and has a proper Readme file.

Reviewer #4

(Remarks to the Author)

Please see the attachment.

(Remarks on code availability)

Version 1:

Reviewer comments:

Reviewer #1

(Remarks to the Author)

After reviewing the revised manuscript "Alternative Stable States in the Human Microbiome," I am satisfied that all eight substantive comments have been fully addressed. In brief, the authors clarified control conditions and timelines (new Fig. 4C), replaced indistinct PCA and bar plots with clearer line + symbol and color-gradient visualizations (Figs. 5 & 6), supplied a comprehensive phenotype–metabolite table, added missing axis labels, and inserted a discussion paragraph (lines 406-416) linking their work to earlier multistability models.

An optional tweak could further polish the paper during production:

Figure 6 y-axis: replot metabolite trajectories on a log scale to match the species panels and reveal low-abundance metabolites.

These are discretionary; the manuscript is publishable as is.

(Remarks on code availability)

Reviewer #3

(Remarks to the Author)

I've reviewed the revised manuscript and I'm pleased to see that all of my prior concerns have been comprehensively addressed or justified by the author's rebuttal letter. In its current form, the paper is much stronger, more transparent, and far better structured.

The authors have executed a thorough update against each of my earlier comments, aligning the manuscript with rigorous standards for clarity, reproducibility, and narrative flow. I am satisfied with the current version and believe it is ready for publication.

(Remarks on code availability)

Reviewer #4

(Remarks to the Author)

Garza et al. have made a substantial effort in this revision, and the overall quality of presentation has improved significantly. The revised manuscript is more clearly written and better structured, with key figures now much easier to follow. I appreciate the authors' detailed responses and the additional replicates provided, which help clarify parts of the experimental and modeling work.

That said, I believe several core conceptual issues still warrant clarification. In particular, the manuscript's central claim of multistability remains difficult to evaluate given (i) potential nutrient divergence between experimental replicates, (ii) the

deterministic behavior of the mechanistic model under fixed environmental conditions, and (iii) a mismatch between the conceptual landscape of multistability (as depicted in Fig. 1) and the model outputs (e.g., Fig. 4A). My goal is not to undermine the substantial effort already made, but to help ensure that the conclusions are as robust and compelling as possible. I outline below five main areas where clarification or additional analysis could strengthen the manuscript.

1. Clarifying true multistability vs. nutrient-driven divergence

While I appreciate the improved clarity and the inclusion of additional replicates, I would like to clarify that my concern was not about whether extrinsic parameters such as pH and dilution rate were restored. Rather, my key question is whether the internal environmental conditions—specifically, nutrient and metabolite concentrations—were equivalent across vessels at the time when divergence in community compositions occurred. Without establishing that vessels stabilized under comparable nutrient composition/profiles, the observed divergence in community composition may be better explained by nutrient heterogeneity, rather than by true multistability.

In this regard, Figure 4C and Figure 5 actually reinforce my concern. The data show that vessels subjected to the same pH + feed perturbation diverged into different community states, but also ended with substantially different nutrient profiles, including clear variation in residual trehalose, glucose, and fermentation products. This undermines the assumption that divergence occurred under identical conditions and strongly suggests that nutrient availability could plausibly drive the resulting community structure.

Furthermore, Figure 6—while helpful in showing time-resolved data for two selected vessels—also supports this interpretation. These two vessels followed distinct compositional trajectories that correlate with divergent nutrient dynamics, particularly in trehalose depletion. In the absence of a broader comparative analysis across all vessels, it remains unclear whether these examples reflect internal dynamics (i.e., phenotype switching) or simply early nutrient divergence followed by deterministic trajectories.

To directly address this ambiguity, the authors should explicitly demonstrate, statistically or visually, whether nutrient conditions at the point of divergence were equivalent. A comparative analysis of nutrient profiles across vessels immediately preceding divergence would clarify causality (species divergence vs. nutrient divergence). Furthermore, I strongly suggest a follow-up re-inoculation experiment in which pre-stabilized communities representing the alternative states are exposed to identical nutrient conditions. If communities maintain their distinct structure under identical nutrient inputs, it will definitively strengthen the claim of multistability. Conversely, convergence under identical nutrients would indicate nutrient-driven determination.

2. Ensuring consistent analyses across experimental setups (Fig. 5 & 6)

I really appreciate the additional replicates being included to Fig. 5 and the tidy up in Fig. 6, but there are some inconsistencies in how these two types of experiments are analyzed. Currently, we have community composition and time-lagged dissimilarity analysis in Fig. 5, whereas Fig. 6 shows the average or individual population dynamics.

To enhance comparability and robustness, it would be extremely helpful if the authors provided equivalent supplementary analyses across the two experimental setups: specifically, (i) population dynamics plots for mini-bioreactor experiments analogous to Fig. 6, and (ii) PCA trajectories, ternary composition plots, and Bray-Curtis dissimilarity analyses for fermenter experiments analogous to Fig. 5. This consistent presentation will allow for clearer interpretation and comparison of community dynamics and strengthen the empirical support for alternative stable states.

3. Resolving conceptual mismatch between proposed multistability and model outcomes

In my previous comment, I raised the concern that it remained unclear whether the model exhibits true multistability — i.e., the existence of multiple stable outcomes under identical external conditions, solely due to differences in prior history. This conceptual issue remains central to the manuscript's interpretation of the experimental data, particularly Figures 1 and 6, which emphasize multistability.

Currently, Figure 4A (bottom panels) attempts to address this via hysteresis simulations. However, the authors have not clarified how the key perturbations (feed+pH) differ fundamentally from single perturbations. Specifically, does a single perturbation (feed-only or pH-only) always return the system to a unique outcome? Or is it exclusively the combination of both perturbations that leads to multiple possible outcomes? Does the range of 'tipping points' broaden with the combination of two perturbations?

Moreover, I note a clear conceptual mismatch between Figure 1 and the actual model outputs shown in Figure 4A. Figure 1 suggests a classic multistability scenario, where the system settles into one of multiple alternative states under identical conditions. However, Figure 4A implies each environmental parameter (dilution rate or pH) maps deterministically to a single steady state, with transitions driven by changes across tipping points. Even the hysteresis simulations explicitly involve a transient environmental change rather than spontaneous divergence under constant conditions.

The authors should explicitly state in the main text whether the model exhibits true multistability (multiple stable states under identical past conditions), or merely displays multistability due to small differences in environmental conditions (dilution rate of 0.039 vs 0.041). If true multistability does exist in the model, explicit simulations clearly demonstrating multiple steady states for identical external conditions should really be illustrated in Fig. 4. If the model remains environmentally deterministic, the authors should clearly acknowledge this limitation and revise the manuscript framing accordingly.

4. Clarifying narrative inconsistencies around multistability

There appears to be an inconsistency in how the authors frame their core concept of multistability between the manuscript text and their response to reviewers. Specifically, in their response to reviewer 4, the authors explicitly argue that alternative stable states emerged under identical external environmental conditions following perturbation:

“After the perturbation, vessels are returned to the original control conditions and allowed to stabilize again for over 80 hours. This design reveals history-dependent responses. The memory effects are visible in how vessels exposed to perturbations converge to distinct steady states, despite their nearly identical conditions.”

This response strongly implies true ecological multistability, defined as the coexistence of multiple stable states under the same external environment, differentiated only by their historical trajectories.

This narrative is consistent with the legend of Fig. 1:

“Fig. 1: Phenotype switching mechanism that leads to multistability. According to our hypothesis, in community contexts, multistability can arise from phenotype switching between subpopulations of the same species. This switching is triggered by small shifts in steady-state metabolite concentrations, which are influenced by environmental factors such as pH and dilution rate. Because the switching is not necessarily reversible in a symmetrical manner, the system may exhibit history-dependent steady states (i.e., hysteresis).”

Nevertheless, the discussion section of the manuscript presents a subtly different scenario, suggesting that alternative states primarily arise from shifts in environmental conditions and phenotype responses, rather than historical contingency alone. For example, the authors state:

“However, in our system, these shifts are triggered by changes in environmental parameters and the associated microbial phenotypes.”

Thus, the manuscript’s narrative inadvertently drifts from strict multistability toward something resembling environmental hysteresis—where environmental conditions (even subtle differences) lead to different outcomes—rather than purely historical contingency under identical conditions.

To resolve this conceptual mismatch clearly, the authors should explicitly clarify in the manuscript whether their observed multistability arises strictly from historical contingencies under truly identical external environmental conditions, or whether subtle environmental variation indeed plays a crucial role. Such explicit clarification would greatly strengthen the manuscript’s conceptual coherence and precision.

5. Parameter fitting

The post hoc RMSE sensitivity plots (Fig. S7) suggest that a subset of fitted parameters are locally well-constrained, with RMSE minima centered near the estimated values. However, many parameters show minimal or no sensitivity across a wide range (0.1x- 10x), indicating they may be poorly informed by the data or structurally non-identifiable. This concern is amplified by the use of Powell’s method, which is a local, derivative-free optimizer that can struggle in high-dimensional parameter spaces.

Additionally, several fitted parameters appear pinned at or near their upper or lower bounds (e.g., ri_xi_mumax , ri_xi_pHopt , $ri_xi_pHalpha$ in `ri.tsv` on GitHub; and many values in Table 2 of Supplementary Text S1 that round to integers like 1.000, 2.999, or 10.000). While some of these parameters show low sensitivity in the RMSE analysis, this does not preclude the possibility that their values reflect optimization constraints rather than genuine biological signals.

Thus, the authors should explicitly acknowledge these parameter-fitting limitations within the main text. Moreover, they should provide the full set of parameter bounds used, clearly justified biologically or computationally (e.g., as Table 3 in Supplementary Text S1). Finally, I recommend conducting a formal parameter identifiability analysis or employing regularization methods to explicitly assess and demonstrate model robustness. Without addressing these concerns, the biological interpretability and predictive power of the model parameters remain uncertain.

(Remarks on code availability)

The authors provide executable Python notebooks for reproducing the main theoretical figures, which I greatly appreciate. However, the parameter fitting pipeline is less transparent. While parameter outputs and value constraints are included in the `params/` folder, no executable code or documentation is provided for the fitting procedure itself. This limits reproducibility of the core modeling results and makes it difficult to evaluate identifiability or optimization behavior.

Version 2:

Reviewer comments:

Reviewer #4

(Remarks to the Author)

Please see the attached PDF for a structured comment and accompanying figure.

(Remarks on code availability)

Thanks for uploading the parameter fitting codes!

Reviewer #5

(Remarks to the Author)

From reading the back and forth between the authors and reviewer #4 I feel comfortable with the conceptual approach being used by the authors. I would note that resource concentrations and pH can be part of the feedback loop leading to multi stability, so the fact that these concentrations are different at the end of the experiment does not mean that the observation is not "true multi stability". However, I do feel that the experiments are not the cleanest demonstration of multi stability. In particular, perturbations can induce transitions between alternative stable states, but they can also lead to extinction of a species and the resulting final state is not necessarily a stable state, in that it is possible that the final state of the community can still be invaded by the species that went extinct during the perturbation. For example, in Figure 5C the authors show that a Ph followed by feed perturbation leads to three possible outcomes: 1) coexistence of all three species, 2) extinction of B. theta and coexistence of the other two, and 3) extinction of two species and survival of B. hydro. I would note that these are not necessarily stable states, since it is possible (likely?) that outcomes (2) and (3) correspond to community states in which one or both of the extinct species could invade, leading the community to end up at state (1). I feel that the authors should at least acknowledge this as a possibility.

(Remarks on code availability)

Reviewer #6

(Remarks to the Author)

.

(Remarks on code availability)

.

Point-by-point response to the reviewers' comments

Reviewer comments are reproduced verbatim with a black font color. Our responses follow with a red font color.

Reviewer #1 (Remarks to the Author):

This manuscript represents a novel and highly significant contribution to the field of microbiome research. The study uniquely integrates metabolomics and species abundance data to investigate alternative stable states in a synthetic human gut microbiome. The experimental validation of multistability alongside the mechanistic modeling provides insights into community-level dynamics.

This work is the first to study alternative stable states in human gut microbiomes with this level of mechanistic detail, measuring both metabolite concentrations and species abundances as a function of chemostat dilution rate and pH of the medium.

While the study is scientifically robust, the clarity of the results presented in Figures 2- 4 is a critical issue. These figures are central to the paper's main message, yet their current presentation makes the results difficult to understand. The modifications recommended below are presented in order of their relative importance.

Thank you for the kind and rigorous assessment of our manuscript. In response to the concerns regarding the clarity of Figures 2–4, we have made several changes to improve how the results are presented:

- We added a new summary Figure (**Fig. 1**), that illustrates our main hypothesis;
- We updated all Figs with clearer labels and legends;
- We split the PCA plots from the old **Fig. 3B** into a new figure (**Fig. 5**) that better shows the trajectory of the experimental validation;
- The old **Fig. 4** (now **Fig. 6**) was also improved for better readability.

Recommendations for Revisions:

1. The experiment presented in Figure 3C is completely unclear. Provide a detailed description of what happened over time and at which specific time points. Clearly define what constitutes the control conditions.

Thank you for the helpful comment. We put considerable effort into improving the presentation of these experiments in the revised manuscript. As mentioned above, we now organize the minibioreactor experimental results across three main figures:

- **Fig. 4C** shows the steady states reached under each condition (control, feed perturbation, and pH/feed perturbation), with clearer axis labels and a description of the control and perturbation setups.
- **Fig 5** presents PCA trajectories over time, supported by ternary plots and a sliding-window Bray–Curtis analysis to illustrate convergence toward steady states.
- **Fig. 6** shows the full time series from one control and one perturbation experiment, highlighting the divergence in trajectories and the emergence of alternative stable states in two independent vessels.

2. The central message of the paper—transitions between alternative stable states—is poorly represented in Figures 3D and 3E. It is essential to show these transitions more clearly. Where are alternative stable states in Figure 3E? Where can I see memory effects in transitions between these states in the experimental part of Figure 3?

Thank you for pointing this out. As we mentioned above, in the revised manuscript, we reorganized and clarified the experimental results across three main figures (**Figs. 4C, 5, and 6**) to better highlight transitions and memory effects:

We now describe the perturbation experiments more clearly in the figure captions, explicitly stating the sequence of conditions. For example, vessels are first stabilized under control conditions (defined in the Fig. legend: 37 °C, uncontrolled pH, and dilution rate $\approx 0.04 \text{ h}^{-1}$) for over 100 hours. They are then subject to perturbations (Feed perturbation: dilution is stopped (set to 0) for 12 h; pH + feed perturbation: acidified medium is introduced for 24 h, followed by 12 h with dilution set to 0). After the perturbation, vessels are returned to the original control conditions and allowed to stabilize again for over 80 hours. This design reveals history-dependent responses. The memory effects are visible in how vessels exposed to perturbations converge to distinct steady states, despite their nearly identical conditions.

3. Using greyscale shadows of different intensity in Figures 3D and 3E is ineffective. Incorporating a gradient in intensity of colors will better depict changes in states over time.

These figures were substantially revised and are now presented as **Fig. 5**. We increased marker sizes and labeled areas of interest so that it is clear which time points are relevant. In our opinion, this worked better than a gradient of colors. We used a purple bar for time that clearly distinguishes early from later time points as you can see in the new Fig:

4. The use of bars to denote species abundances in three-species communities in Figure 4 is ineffective. These bars are too small to discern differences in species composition. I recommend switching to a log-scale for the Y-axis to better represent species abundances and use line+symbols plots similar to those used for control panels.

Thank you for the helpful suggestion. We implemented these changes in the revised Fig. 6.

5. Figure 2A, which illustrates phenotype transitions and metabolic interactions, is difficult to interpret due to the "spaghetti-like" lines connecting species and metabolites. It is nearly impossible to trace how different phenotypes transition and how these transitions modify which metabolites are consumed or produced by each of three species. A more effective alternative might be to present this information as a table listing each phenotypic state along with the metabolites each species consumes and produces. This would provide a clearer view of the key interactions driving the alternative stable states.

We added a table with this information as suggested (Table1)

6. The paper reports phenotypic transitions in metabolic utilization and production of metabolic byproducts. These include but are not limited to diauxic shifts, e.g. the switch from trehalose to glucose utilization by *Blautia hydrogenotrophica*. Diauxic shifts were responsible for alternative stable states in a stylized model described in Ref. 8. Highlighting the connection and differences between the results of this study and existing models of alternative stable states in human gut microbiome in the discussion would strengthen the theoretical foundation of the study and contextualize these findings within broader ecological frameworks.

Thank you — we agree that this connection strengthens the theoretical context of our work. Several mechanisms other than environmental impact were suggested in the literature to explain alternative stable states in the gut microbiome. Gibson et al. (ref. 6) combined a strongly uneven distribution of interaction strengths with sub-sampling from a species pool to generate alternative stable states. Biologically speaking, this means that a priority effect (i.e. the initial presence or absence of a species with strong interactions) determines community composition. This mechanism assumes different local community compositions and thus cannot explain alternative stable states observed after a perturbation. Another mechanism, proposed by Goyal et al. (ref. 8), links microbes to nutrients based on nutrient preferences and competitive ability using the stable marriage model. This model was applied to predict alternative stable states among *Bacteroides* species growing on different polysaccharides. According to Thomas' conjecture, a positive feedback loop and non-linearity are necessary conditions of multi-stability. We previously used these ingredients to build a toy model for microbial communities that displayed alternative steady states upon perturbation without strongly heterogeneous interaction strengths (see ref. 7).

Conceptually, the stable marriage model comes closest to the mechanism postulated here. However, it is a qualitative model that only makes rank-based predictions. Our detailed mechanistic model also considers aspects that the stable marriage model does not tackle, specifically the effects of metabolite production or consumption beyond competition for nutrients. For instance, cross-feeding of metabolites such as acetate and changes in pH resulting from short-chain fatty acid production also impact the dynamics.

These considerations are now summarized in the discussion (lines 406-416).

7. Panels in Figure 4 lack labels, making interpretation challenging. Add clear labels to each panel and use distinct colors for metabolites and species to prevent confusion.

Done

8. Figure 3C is missing an X-axis label, and Figure 2C is missing a Y-axis label. Ensure that all figures have properly labeled axes to improve clarity.

Labels were added.

Conclusion: This manuscript is a groundbreaking study that advances the understanding of microbial community stability and dynamics. The findings are highly significant, but the presentation of key results requires substantial improvement. Addressing these issues will greatly enhance the accessibility and impact of the paper. I strongly recommend acceptance after these revisions.

Thank you for the encouraging assessment. We've carefully revised the figures and presentation of key results to improve clarity and accessibility, and we hope the changes meet your expectations.

Reviewer #3 (Remarks to the Author):

Garza et al., the authors demonstrated that phenotype switching between subpopulations within the same species is responsible for multi-stability in bacterial communities. They first explored the life history strategies of three gut bacterial species using RNA sequencing and metabolomics in monoculture. Next, they developed a parameterized kinetic model to predict community structure. Finally, they utilized a simple three-species consortium in a mini-bioreactor to validate their findings on phenotype switching and alternative stable states.

Although the study is quite informative, appealing, and worthy of publication in Nature Communication, the authors need to clarify certain sections of the article. My comments are detailed below, organized by section of the paper.

Thank you for the encouraging assessment of our work. We've addressed your comments point by point below, and we hope the revisions clarify the manuscript and fully resolve your concerns.

Abstract & Introduction:

The author's should rephrase "Thus, a transient perturbation combined with metabolic flexibility is sufficient for alternative communities to emerge, implying that they are not necessarily explained by differences between individuals." As the sentence itself implies, the difference in individuals has nothing to do with community composition, which is not demonstrated in the paper. I recommend rephrasing this sentence.

Thank you for pointing this out. We agree that the sentence could be misinterpreted and thus we removed it from the abstract. The idea we originally intended to convey is stated more clearly in the discussion: alternative community types may arise from past transient perturbations rather than from persistent differences between hosts, such as diet or genetics.

Most of the introduction follows a chronological order; I would still like to see a summary paragraph at the end. The introduction concludes by stating the hypothesis, which seems somewhat unusual.

We added a short summary of our findings to the conclusion of the introduction as requested by the reviewer (lines 71-76).

Result 1 (Life history strategies of gut bacteria in a heterogeneous environment):

- Is there a rationale for selecting these three specific bacterial species? Was the selection random, or was it based on the bacteria's ability to consume distinct nutrients? A brief explanation would be helpful.

Thank you for the suggestion, we added a sentence clarifying our rationale for species selection: these three species represent a significant fraction of the adult gut microbiome, occupy distinct ecological niches, and can be reliably cultured in our culture medium (lines 79–82).

- For Fig. 1A, why are there two panels labeled 14h/32h? Did you mean 14h/72h?

This was a typo and has been corrected in the revised figure (now Fig. 2).

- The figure legends should be more informative. They need to specify the number of replicates (for example, by including numbers in brackets) and clearly explain the significance of the value cutoffs and any statistical corrections used. This suggestion applies to all figures in the manuscript.

- It would also be useful if the authors provided a summary of the RNA-seq data. Are there additional differentially expressed genes? What percentage of these genes are related to nutrient utilization beyond the predicted targets? While the focus is on the expected subset of genes, the current text implies that only those genes are significant. A supplementary panel summarizing these data would improve clarity.

We updated the figure legends to include sample sizes, significance cutoffs, and details on statistical corrections. We also added a full summary of the RNA-seq analysis in Supplementary Fig S1 and Supplementary Table S1, showing all metabolic genes significantly associated with specific growth stages, including those beyond our predicted targets.

- How confident are the authors that no other nutrient utilization occurs in monoculture? Although untargeted metabolomics does not provide complete annotation, it can still demonstrate that the predicted sugars drive growth.

We previously performed metabolomics in WC medium and detected trace amounts of several sugars, including galactose, fucose, and mannose. While we can't rule out minor contributions from these compounds, our kinetic modeling and experimental results are consistent with the compounds we were able to measure directly in the current study.

In parallel work (see Ref. 24), we also tested the growth of *Bacteroides thetaiotaomicron* and *Roseburia intestinalis* in WC medium supplemented with fucose, fructose, and mannose. We further explored the role of these sugars for *Blautia hydrogenotrophica* as well.

In WC medium without yeast extract, we found that supplementing with trehalose led to clearly higher growth compared to fucose, fructose, or mannose, both in the presence and absence of yeast extract:

Group A: no pyruvate, glucose, and yeast extract

Medium composition (in Con)	g/L
Tryptone	10
L-Arginine	1
Gelatin Peptone	10
Sodium Pyruvate	0
Yeast Extract	0
Hemin	0.005
Sodium Chloride	5
Vitamin K3	0.0005
Glucose	0

The sugars were added at a concentration of 2g/L to each treatment group.

Group B: no pyruvate, glucose. 5 g/L yeast extract

Medium composition (in Con)	g/L
Tryptone	10
L-Arginine	1
Gelatin Peptone	10
Sodium Pyruvate	0
Yeast Extract	5
Hemin	0.005
Sodium Chloride	5
Vitamin K3	0.0005
Glucose	0

The sugar was added at a concentration of 2g/L to each treatment group.

The highest OD value was found here, confirming that trehalose and an appropriate amount of yeast extract (consistent with the original WC medium) promote the growth of Bh.

- In Supplementary Movie S1, the authors claim bimodal growth. While I agree that two distinct subpopulations are present, how do the authors justify the statement regarding “the presence of similar subpopulation sizes of non-dividing trehalose consumers and dividing glucose consumers”? It appears that after the initial bimodal separation, the subpopulations continue to divide, as indicated by an increase in density. Plotting the number of cells in the two separate clusters after the separation would provide clearer evidence.

This point was not clearly explained in the original text, and we clarified it in the revised version (lines 115–121). The measurements in Supplementary Movie S1 are based on SYBR Green and propidium iodide (PI) staining, both of which target DNA. Since *Blautia hydrogenotrophica* cells have the same genome sizes, we interpret the appearance of a subpopulation with weaker DNA staining—observed at the time of the switch from trehalose—as indicative of a fraction of cells that are not dividing.

Result 2 (The model’s stability landscape reveals sharp transition zones):

- Figure 2A is unclear because the central cylinders are shown in multiple colors. Do these colors indicate specific attributes? It appears from legend that two of the cylinders correspond to one bacterium. Consider making this explicit by labeling the bacteria with corresponding colors on the cylinders. The figure legend also needs to be more detailed and clearer.

We thank the reviewer for the helpful suggestions. We have now explicitly labeled the cylinders and added a table to provide clearer information about the links between phenotypes and metabolites. The figure legend has also been revised to improve clarity.

- For Figure 2B, please include a labeled legend for the gradient colors, similar to the other figures.

We added labels to the legend to clarify the color gradient.

- Overall, the results are well presented up to Figure 2B. However, the sudden introduction of the bioreactor systems later in the results disrupts the narrative flow. It is recommended to either move the bioreactor figure to a later section or provide a clearer explanation of the experiment.

As noted in our response to Reviewer 1, we restructured the presentation of the minibioreactor results to improve clarity and narrative flow. These experiments are now organized across three separate figures (Fig. 4C, 5, and 6), with revised captions and legends that explain the design and results more clearly.

Result 3 (Phenotype-switching explains multistability in silico and agrees with observations in vitro)

- For Figure 3, the lower inset requires clearer annotation; the authors should label this inset as Fig. 3C. The manuscript states that hysteresis influences community composition, but it is unclear whether this effect is driven by the initial inoculum concentration or by the metabolic memory of the cells. A bioreactor experiment using an inoculum that mimics post-perturbation conditions could help clarify this issue and might also explain the three distinct steady states observed during pH perturbation. Additionally, highlighting or circling the two replicate clusters in the PCA would enhance clarity

This is an important and interesting point, but addressing it experimentally would require substantial additional work beyond the current study. In our model, metabolic memory is captured through the relative abundances of phenotypically distinct subpopulations. However, controlling these subpopulations in vitro is currently not feasible. It would require both (i) the ability to detect phenotypes within mixed communities—currently beyond our experimental abilities—and (ii) a way to fix cells in a given phenotype prior to inoculation. The latter may be achievable with engineered genetic switches, but neither *Blautia hydrogenotrophica* nor *Roseburia intestinalis* are genetically tractable at this point.

Simply adjusting species abundances is not sufficient: in our experiments, we observe more variability in post-inoculation abundances across control vessels than across perturbed vessels, suggesting that the distinct outcomes we see following perturbation are not due to initial abundances but rather to the effect of the perturbation itself.

While our model proposes phenotype switching as a mechanism for multistability, we are not able to directly observe this switching at the community level as we clearly state in the discussion (lines 403–405).

- Regarding Figures 4A and 4B, while these are mentioned in the text, the figures themselves lack adequate annotation. The authors should consider labeling the panels sequentially (e.g., from Figure 4A to 4L) and using these labels consistently throughout the results section.

We substantially revised this figure also in line with reviewer 1's recommendations. In the main text, we now highlight only one representative experiment (**Fig. 6**) and have applied consistent labeling and color coding across all panels to improve clarity and readability.

- The rationale for including a 4-species community should be more explicitly detailed in the results section. The authors need to explain why this community was chosen and what motivated the experiment.

Including a four-species community supports the generality of our findings by showing that multistability is not limited to a specific three-species configuration. However, we were unable to parameterize a four-species model because we could not consistently obtain successful monocultures of *Faecalibacterium duncaniae* in WC medium.

- Figure S5 is not referenced anywhere in the results. The authors should cite this figure in the text.

Figure S5 was replaced by Figure S6 and is now referenced in the text (lines 306, 307, 332, 345)

Reviewer #3 (Remarks on code availability):

The Github page is well-organized and has a proper Readme file.

We also revised the GitHub page to include a section with notebooks to reproduce our key model simulations:

<https://github.com/danielriosgarza/hungerGamesModel/tree/main/notebooks>

Reviewer #4 (Remarks to the Author):

Paper summary

The paper titled “discovery of alternative stable states in a synthetic human gut microbial community” utilizes three common human gut bacterial species to form simplified communities and analyzes the community composition under various environmental conditions and perturbations. The work combines experiment data and ODE simulations and suggest multiple steady states can emerge from the same community. In particular, the ODE simulations were generated from an elaborate community modelling framework, as the authors tried to provide a full metabolic pathway of the community. Moreover, on the empirical side, the authors provide a nice exploratory dataset including both population and nutrient dynamics with good temporal resolutions. However, there are multiple fundamental issues that led us to doubt the suitability of publishing in Nature Communications, as described in the comments below:

We appreciate the reviewer’s feedback and have carefully revised the manuscript in an effort to respond to all issues listed below.

General comments

1. The main claim “finding the alternative stable states” is ill-supported by the experimental data. Firstly, no statistics were used to support the notion of alternative stable states, and we cannot tell if a distribution is truly bimodal from just two samples (bioreactor vessels). From the graphs, it looks like a range of shifting community states in response to but even in the absence of a perturbation.

We have thoroughly revised the presentation of the minibioreactor experiments to better support the claim of alternative stable states (also see replies to reviewer 1 and 2). In the revised manuscript, we first show the endpoints of 24 independent replicates cultured for 84 hours under control conditions (37 °C, uncontrolled pH, and a dilution rate of 0.04 h⁻¹). Despite some experimental noise—expected in anaerobic gut bacterial cultivation—the data show a consistent convergence to a common community structure.

We then present six replicates subjected to a feed perturbation (dilution rate set to 0 for 12 h), followed by restoration to control conditions. After 90 hours, all vessels converged to the same community state, distinct from the control group, with *Roseburia intestinalis* consistently outcompeted and minimal variation across replicates.

Finally, in the pH + feed perturbation, also followed by restoration to control conditions, the six replicates diverged into three clearly distinguishable final community states, again measured after 90 hours.

To support these findings, **Fig. 5** now shows the full trajectories for all replicates. In the control condition, all trajectories converge to a single region in PCA space, with composition plots confirming consistent coexistence of the three species (with only four potential measurement outliers). In the feed perturbation, trajectories shift away from the control cluster and converge tightly to a new state lacking *R. intestinalis*. In the pH + feed perturbation, replicates diverge into three distinct clusters in both PCA space and species composition plots, consistent with the emergence of alternative stable states.

In an alternative visualization, **Fig. 6** displays abundance and metabolite concentration time series for control and selected perturbed vessels, clearly illustrating that the community composition in control vessels converges to one stable state, whereas vessels exposed to the same transient perturbations reach different end states.

In total, three separate perturbation experiments were performed, the first consisting of six perturbed vessels, the second of six perturbed and 12 control vessels (both summarised in **Fig. 5** and Supplementary **Fig. 6**), and the third consisting of six perturbed and 12 (8 processed) control vessels (shown in **Fig. 6**). In all three perturbation experiments, alternative stable states emerged.

In our opinion, these data show repeatable, condition-dependent divergence into discrete states and provide strong experimental support for multistability in this system.

Secondly, another issue is that the data are inconclusive as to whether those states are stable, as there was no statistical analysis to support the claim on stability.

The experiments were conducted in mini-bioreactors that emulate chemostat conditions, and our assessment of stability is based on two main observations:

- (i) the convergence of biological replicates to consistent species and metabolite profiles within each condition (see revised **Fig. 4C** and **Fig 5**), and
- (ii) reduced variability between time points as the system evolves.

To support this, we added new plots showing sliding-window Bray–Curtis dissimilarity between consecutive time points. In all three experimental conditions, we observe that divergence between time points drops over time and remains low in the final phase of cultivation. This pattern indicates that the communities reach stable states around the selected time windows.

Moreover, our third main criticism is of conceptual nature: The authors are not finding “alternative” stable states because the community does not

converge on multiple alternative compositions under the exact same environmental conditions. Changes in community in response to environmental changes are hardly a novel finding, as they directly follow from the idea that bacteria adjust their gene expression to environmental changes (e.g. following by the seminal work of Jacob and Monod). If the message of this manuscript amounts to “a community changes in response to a different environment” instead of finding alternative stable states seriously undermine the novelty of this manuscript. In other words, the fact that a shift in diet changes the microbiome’s composition is not novel.

We respectfully disagree with the reviewer’s interpretation. Both our model and experiments provide evidence for true multistability, not just a shift in community composition due to changing environmental conditions.

The key point is that the observed community states persist even after the perturbation ends and the system is restored to its original environmental conditions. This is fundamentally different from the well-known and expected response of bacteria to current environmental cues (as described by Jacob and Monod). In our study, the same environment supports more than one, stable community state depending on prior history.

In the model, this is reflected in regions of parameter space where multiple steady states are possible (see Supplementary Fig. S5). Experimentally, we show that after transient feed or pH + feed perturbations, the system converges to distinct and reproducible endpoints, despite being returned to the original environmental conditions. This persistence is the signature of multistability.

We acknowledge the limitations of our current setup—for example, we are not yet able to construct inocula that replicate post-perturbation phenotypically different subpopulation structure (see answer to reviewer 2)—but within those constraints, the evidence supports history-dependent divergence, not just environmentally-driven shifts.

2. The main text failed to highlight the key messages; instead, it is a series of descriptions of metabolic phenomena/behaviors and lacks clarity which statement is derived of modeling versus experimental results. A concise graphical abstract showcasing the most important metabolic functions/mechanisms that are supposed to give rise to the predicted alternative steady states would help.

We have added a new figure in the main text that highlights the central hypothesis of the study, as well as Supplementary **Fig. S4**, which illustrates how phenotype switching underlies multistability in our model. We also improved the Figures with minibioreactor experiments (specifically, splitting the multicomponent plot into an independent Figure (**Fig. 5**) that better clarifies the statements derived from experiments and from the model.

2. The quality presentation needs to be improved to be understandable to general microbiologist audiences. For instance, in Fig. 4, the pH was not in the legend, and populations are presented in a cumulative style with very thin columns, which makes understanding the absolute abundance literally impossible. It is also unclear why the panel on the left-hand side shows only such sparse sequencing data in comparison to the right-hand side panel. Similarly, the differences between experiments in Fig. 1 is not clear (graphically). In addition, the bioreactor system figure in Fig. 2C is misleading because the coexisting graph is purely computational, not empirical.

We revised Fig. 4 (now Fig. 6) to improve readability by replacing the stacked bar plots with clearer line + symbol plots, and we added pH information and sample sizes to the legend. In addition, the legend of Fig. 2C (now Fig 3C) has been updated to make it clear that the coexistence diagram is derived from model simulations, not experimental data.

Lastly, the meaning of columns in Fig. 3C is unclear but different from Fig. 3A and B (which is the dilution rate); are they observations from different time points?

This has been clarified in the revised figure (now Fig. 4C) and its legend. The meanings of the columns are now explicitly described to avoid confusion. This point was also addressed in our responses to other reviewers.

4. The model fitting process is only described with one sentence at the end of supplementary document. This is a huge problem considering there are more than 100 parameters in the full model, where fitting values is a very challenging task. The authors mentioned the loss function and the optimization algorithm, but completely skipped explaining the fitting routine, such as which variable was fitted first and how the nutrient data was processed before fitting. The online code repository has the very same problem, that there are codes for moving average and splines but no codes for the optimization process. In addition, there is no confidence or credible intervals of the fitted values, not to mention that the overall structure of the supplementary is sloppy (e.g., sections are not properly indexed and hard to navigate).

Thank you for identifying these problems. We have carefully revised Supplementary Text S1 to improve both its organization and clarity. The section on model fitting has been significantly expanded to describe the full optimization process, including the variables fitted, data preprocessing steps, and rationale for the chosen method.

We now explain that the fitting procedure was based on minimizing a pseudo-Huber loss function computed across all measured variables (cell counts, pH, and extracellular metabolites), using spline-smoothed trajectories of the experimental data. Parameter optimization was carried out using Powell's method, a derivative-free algorithm suitable for high-dimensional, non-convex landscapes.

We also include a step-by-step summary of the fitting workflow, from moving average computation and spline interpolation to simultaneous loss minimization. While the optimization method used (via the `lmfit` package) does not provide statistical confidence intervals, we performed a post hoc sensitivity analysis by varying each fitted parameter across a broad range (0.1× to 10× its estimated value) and computed the RMSE to the experimental data. This analysis is now presented in Supplementary **Fig. S7**.

5. Model design is a bit under-justified. The authors used the metabolic pathway supported by RNA-seq data as the backbone of the entire model. Nevertheless, this approach does not exclude the possibility of additional pathways being omitted, which is important considering the communities grew in a rich medium. Similarly, it seems the authors put no effort to reduce the number of variables in the model. For example, why do we need an inactive and a dead population? If there is a close to zero probability for those cells to reactivate, then the dead population is redundant.

The model was designed to reflect measurable features of the system, following a pragmatic approach rooted in empirical observations from monoculture experiments. While we recognize that this is a simplification and does not capture all possible metabolic pathways—particularly given the complexity of growth in rich medium—it was not our aim to construct a fully comprehensive mechanistic model of the system. Rather, our goal was to capture the dominant metabolic features we could observe, validate with RNA-seq and genome-scale metabolic reconstructions, and use this to explain the dynamics seen in co-culture. At present, there is no established method that accurately derives a full kinetic model from high-throughput sequencing or omics data alone—this remains an open and important challenge for the field, and a topic we are actively exploring in ongoing work.

We agree that model reduction is an important consideration, but it was beyond the scope of the current study. Nonetheless, we do present a simplified phenomenological model that illustrates the core mechanism—phenotype switching—leading to alternative stable states.

Regarding the inclusion of inactive and dead populations: this distinction was based on experimental observations. Specifically, we found that the clearance rates of inactive cells differed across species, and including an intermediate "inactive" state allowed us to fit the data from inactive cells in the model (to see the how this state is impacted by model parameters, refer to Supplementary **Fig. S7**).

In summary, some sensitive analysis should be considered to show the findings are robust against adding or removing some metabolic pathways.

Metabolic pathways were not explicitly included in the model. While artificially adding or removing additional carbon sources could be explored in future studies, we believe this would not strengthen or weaken the conclusions presented here. The model was designed to reflect what we could measure directly, rather than to simulate hypothetical pathways.

To address robustness more directly, we added two forms of sensitivity analysis:

(i) Supplementary **Fig. S4**, which shows that the emergence of multistability in the model depends on the transitions between subpopulations, and (ii) the parameter sensitivity analysis described above, in which we varied parameter values across a broad range and tracked their influence on model fit (**Fig. S7**).

Specific comments - Author contributions: it is unclear who did the experiments.

This was added in the revised manuscript.

- The PCA in Fig.3 is hard to understand and a 3D figure did not help.

This Figure was fully revised and now presented as Fig. 5 (see above)

- Why there is only two bioreactor data being shown while the device can run more? Was there data being thrown away?

This plot was revised and is now **Fig. 6**. We show the average of multiple vessels and highlight two cases that show convergence to alternative states.

- Consider renaming some variables to improve readability. For instance, change $\mu_{max,c}$ to $\mu_{max,c}$, and pH_{opt} to pH_{opt} .

The variables were carefully revised for a greater consistency (see Supplementary **Text S1**).

- Add figure and table numbers to the supplementary document.

done

- Parameter table in the supplementary document: make sure the values are in the same format. Some 10 have different numbers of digit after the decimal place

All parameters now have 10 digits.

- Supplementary document page 5, before the figure with Ma and Mb: a paragraph is duplicated.

As mentioned above, the full Supplementary Text S1 was revised. This and other inconsistencies were fixed.

REVIEWER COMMENTS

Reviewer #1 (Remarks to the Author):

After reviewing the revised manuscript “Alternative Stable States in the Human Microbiome,” I am satisfied that all eight substantive comments have been fully addressed. In brief, the authors clarified control conditions and timelines (new Fig. 4C), replaced indistinct PCA and bar plots with clearer line + symbol and color-gradient visualizations (Figs. 5 & 6), supplied a comprehensive phenotype–metabolite table, added missing axis labels, and inserted a discussion paragraph (lines 406-416) linking their work to earlier multistability models.

An optional tweak could further polish the paper during production:

Figure 6 y-axis: replot metabolite trajectories on a log scale to match the species panels and reveal low-abundance metabolites.

These are discretionary; the manuscript is publishable as is.

We appreciate the positive response of the reviewer. Concerning Figure 6, we opted not to employ a log scale for metabolite concentrations to illustrate more clearly their divergence in communities reaching different stable states.

Reviewer #3 (Remarks to the Author):

I’ve reviewed the revised manuscript and I’m pleased to see that all of my prior concerns have been comprehensively addressed or justified by the author’s rebuttal letter. In its current form, the paper is much stronger, more transparent, and far better structured.

The authors have executed a thorough update against each of my earlier comments, aligning the manuscript with rigorous standards for clarity, reproducibility, and narrative flow. I am satisfied with the current version and believe it is ready for publication.

We thank the reviewer for this positive response.

Reviewer #4 (Remarks to the Author):

Garza et al. have made a substantial effort in this revision, and the overall quality of presentation has improved significantly. The revised manuscript is more clearly written and better structured, with key figures now much easier to follow. I appreciate the authors' detailed responses and the additional replicates provided, which help clarify parts of the experimental and modeling work.

That said, I believe several core conceptual issues still warrant clarification. In particular, the manuscript's central claim of multistability remains difficult to evaluate given (i) potential nutrient divergence between experimental replicates, (ii) the deterministic behavior of the mechanistic model under fixed environmental conditions, and (iii) a mismatch between the conceptual landscape of multistability (as depicted in Fig. 1) and the model outputs (e.g., Fig. 4A). My goal is not to undermine the substantial effort already made, but to help ensure that the conclusions are as robust and compelling as possible. I outline below five main areas where clarification or additional analysis could strengthen the manuscript.

1. Clarifying true multistability vs. nutrient-driven divergence

While I appreciate the improved clarity and the inclusion of additional replicates, I would like to clarify that my concern was not about whether extrinsic parameters such as pH and dilution rate were restored. Rather, my key question is whether the internal environmental conditions—specifically, nutrient and metabolite concentrations—were equivalent across vessels at the time when divergence in community compositions occurred. Without establishing that vessels stabilized under comparable nutrient composition/profiles, the observed divergence in community composition may be better explained by nutrient heterogeneity, rather than by true multistability.

In this regard, Figure 4C and Figure 5 actually reinforce my concern. The data show that vessels subjected to the same pH + feed perturbation diverged into different community states, but also ended with substantially different nutrient profiles, including clear variation in residual trehalose, glucose, and fermentation products. This undermines the assumption that divergence occurred under identical conditions and strongly suggests that nutrient availability could plausibly drive the resulting community structure.

Furthermore, Figure 6—while helpful in showing time-resolved data for two selected vessels—also supports this interpretation. These two vessels followed distinct compositional trajectories that correlate with divergent nutrient dynamics, particularly in trehalose depletion. In the absence of a broader comparative analysis across all vessels, it remains unclear whether these examples reflect internal dynamics (i.e.,

phenotype switching) or simply early nutrient divergence followed by deterministic trajectories.

To directly address this ambiguity, the authors should explicitly demonstrate, statistically or visually, whether nutrient conditions at the point of divergence were equivalent. A comparative analysis of nutrient profiles across vessels immediately preceding divergence would clarify causality (species divergence vs. nutrient divergence). Furthermore, I strongly suggest a follow-up re-inoculation experiment in which pre-stabilized communities representing the alternative states are exposed to identical nutrient conditions. If communities maintain their distinct structure under identical nutrient inputs, it will definitively strengthen the claim of multistability. Conversely, convergence under identical nutrients would indicate nutrient-driven determination.

Thank you for your detailed comments and for allowing us to explain the important issue of distinguishing true multistability from nutrient-driven divergence. We would like to clarify a fundamental point about our experimental design that directly addresses this concern: we are operating a chemostat-like system. Outside the pH perturbation experiments, where the pH of the feed is lowered, all other conditions use the same feed composition as the WC medium. The observed nutrient differences may initially appear to support nutrient-driven divergence; however, steady states in the chemostat context provide a different interpretation. We have added this information to additional parts of the text and figure legends (see the manuscript with tracked changes); previously, it was only mentioned in the Methods and briefly in the Results.

In a chemostat, the nutrient input composition and the dilution rate are held constant across all vessels. In our conditions, the entire volume of the reactor is cycled every 24 hours and is supplied from the same bottle to the minibioreactor vessels. Hence, vessels have a uniform supply of WC medium. Because the inflow medium—and therefore the availability of carbon sources, nitrogen, and other nutrients—is the same for all replicates throughout the experiment, differences in residual metabolite concentrations at the steady state do not reflect differences in nutrient input, but rather differences in community-level metabolic processing.

This is an important distinction: under chemostat conditions, residual metabolites are an outcome of community structure, not a driver. Because the feed is constant and all vessels are continuously diluted at the same rate, the system effectively fixes the external nutrient supply. Therefore, the observed differences in residual trehalose, glucose, and fermentation products are best explained as resulting from alternative steady states processing the same input differently, which is what defines multistability in a chemostat context.

We appreciate your suggestion of a re-inoculation experiment under identical nutrient inputs. But in effect, our chemostat approach already implements this by design. All replicates experience the same external environment continuously. The observed divergence under these conditions provides strong evidence of internal system-level dynamics rather than external nutrient heterogeneity.

This is further emphasized by the control vessels (12 and 6 in two independent experiments), which reach the same stable state in each experiment, illustrating that minor variations in initial abundances and metabolite concentrations across replicates, due to experimental constraints, are insufficient to induce alternative stable states in this system.

Finally, when trajectories diverge, the intrinsic metabolite concentrations reflect each community's response to identical inputs, influenced by internal factors such as starting subpopulations and metabolic switching, as well as their interactions with the constant metabolic inputs. These parameters are complicated to control experimentally. Even if we attempted to create identical starting conditions by adjusting measurable metabolites before inoculation, the system would require a startup phase without dilution to prevent washout. During this phase, microbes alter their environment and the structure of their subpopulations. This limitation is why chemostat operation remains the best available method for testing our hypothesis, given the current inability to control metabolic switching (as we noted in the previous review and the manuscript's discussion).

2. Ensuring consistent analyses across experimental setups (Fig. 5 & 6)

I really appreciate the additional replicates being included to Fig. 5 and the tidy up in Fig. 6, but there are some inconsistencies in how these two types of experiments are analyzed. Currently, we have community composition and time-lagged dissimilarity analysis in Fig. 5, whereas Fig. 6 shows the average or individual population dynamics.

To enhance comparability and robustness, it would be extremely helpful if the authors provided equivalent supplementary analyses across the two experimental setups: specifically, (i) population dynamics plots for mini-bioreactor experiments analogous to Fig. 6, and (ii) PCA trajectories, ternary composition plots, and Bray-Curtis dissimilarity analyses for fermenter experiments analogous to Fig. 5. This consistent presentation will allow for clearer interpretation and comparison of community dynamics and strengthen the empirical support for alternative stable states.

Thank you for the suggestion, we added a supplementary Figure S7, which shows the PCA, ternary plots, and Bray-Curtis dissimilarity for the experiment in Fig. 6.

The population dynamics for the other two mini-bioreactor experiments are shown in Supplementary Figure S6. We also expanded the legend of S6 to clarify which data sets are displayed.

3. Resolving conceptual mismatch between proposed multistability and model outcomes
In my previous comment, I raised the concern that it remained unclear whether the model exhibits true multistability — i.e., the existence of multiple stable outcomes under identical external conditions, solely due to differences in prior history. This conceptual issue remains central to the manuscript's interpretation of the experimental data, particularly Figures 1 and 6, which emphasize multistability.

Currently, Figure 4A (bottom panels) attempts to address this via hysteresis simulations. However, the authors have not clarified how the key perturbations (feed+pH) differ fundamentally from single perturbations. Specifically, does a single perturbation (feed-only or pH-only) always return the system to a unique outcome? Or is it exclusively the combination of both perturbations that leads to multiple possible outcomes? Does the range of 'tipping points' broaden with the combination of two perturbations?

In the manuscript, we show that a feed perturbation, when applied alone, exhibits evidence of true multistability in both the model and experiments (see the discussion below regarding the concept of true multistability). We also demonstrate that the model exhibits multistability in the pH axis. Comparing the simulation of pH and the experiments is not straightforward because, in the minibioreactor system, it is not possible to precisely control the pH at low values. Therefore, we introduced a perturbation by feeding an acidified medium. In the model, this is not exactly what happens when we fix a pH value. In this case, we simply stop estimating the pH from the fermentation acids, keeping it fixed at a defined value. Because such comparison is not easily done, in the manuscript we are careful not to claim to have reproduced the pH conditions that we found in the model. We only claim to have a similar result for the dilution rate and that we observe evidence of multistability both in vitro and in silico. It is an interesting question whether combining perturbations broadens the parameter range within which multistability can occur, but it is out of scope for our work.

Moreover, I note a clear conceptual mismatch between Figure 1 and the actual model outputs shown in Figure 4A. Figure 1 suggests a classic multistability scenario, where the system settles into one of multiple alternative states under identical conditions. However, Figure 4A implies each environmental parameter (dilution rate or pH) maps deterministically to a single steady state, with transitions driven by changes across tipping points. Even the hysteresis simulations explicitly involve a transient environmental change rather than spontaneous divergence under constant conditions.

The authors should explicitly state in the main text whether the model exhibits true multistability (multiple stable states under identical past conditions), or merely displays multistability due to small differences in environmental conditions (dilution rate of 0.039 vs 0.041). If true multistability does exist in the model, explicit simulations clearly demonstrating multiple steady states for identical external conditions should really be illustrated in Fig. 4. If the model remains environmentally deterministic, the authors should clearly acknowledge this limitation and revise the manuscript framing accordingly.

Thank you for raising this point. We recognize that this is a subtle concept, and we appreciate the opportunity to clarify it. We have made small textual revisions to the manuscript to clarify the concept.

Our model, as described in Supplementary Text S1, is a deterministic system of ordinary differential equations (ODEs) built from standard biological components: dilution and inflow terms, microbial growth following Monod kinetics, and phenotype transitions modeled using Hill functions. These functions are smooth and well-behaved across the biologically relevant ranges, ensuring predictable system behavior under fixed conditions.

This property is formalized by the Picard–Lindelöf uniqueness theorem, which states that if a system satisfies certain smoothness conditions (technically, a Lipschitz condition), then for any given set of parameters and initial conditions, there is exactly one possible trajectory. In practical terms, two trajectories starting from the same initial state under the same external conditions will always follow the same path and end in the same state. Spontaneous divergence is therefore mathematically impossible unless variability on kinetic parameters or stochasticity is introduced—neither of which is present in our model, nor did we observe spontaneous divergence in the control vessels.

Importantly, this is not unique to our work. The same property holds for virtually all classical deterministic ecological models that exhibit multistability, including those discussed by May (ref. 9) and by Marten Scheffer in *Catastrophic shifts in ecosystems* (*Nature*, see Box 1). These models also show hysteresis and alternative stable states while obeying the same uniqueness principle: trajectories are deterministic, but the system contains multiple basins of attraction under identical external conditions. As in these studies, hysteresis provides a standard way to demonstrate the existence of such alternative states.

True multistability does not mean that identical trajectories split apart. Instead, it means that under the same external conditions (e.g., feed, dilution rate, and pH), the system can admit multiple stable states, and which one is reached depends on its starting point or history (such as prior perturbations). This is exactly what our hysteresis simulations in Fig. 4 illustrate: once the system crosses a tipping point and then returns to its original external conditions, it remains in the new state rather than reverting to its previous state. This persistence is only possible if more than one attractor exists under the same environmental parameters—a defining feature of multistability. This behavior is not trivial: as shown in Supplementary Fig. S4, it only occurs in our model because of phenotype switching. When we remove phenotype switching, the model returns to its previous state following a perturbation.

Finally, Fig. 4A does not contradict this interpretation. It shows the state reached from a single initial condition for each parameter value. Such representations cannot reveal coexisting attractors unless multiple starting points are tested under the same conditions, which we illustrate in the insets of Fig. 4A–B and in Supplementary Figs. S4 and S5.

4. Clarifying narrative inconsistencies around multistability

There appears to be an inconsistency in how the authors frame their core concept of multistability between the manuscript text and their response to reviewers. Specifically, in their response to reviewer 4, the authors explicitly argue that alternative stable states emerged under identical external environmental conditions following perturbation:

“After the perturbation, vessels are returned to the original control conditions and allowed to stabilize again for over 80 hours. This design reveals history-dependent responses. The memory effects are visible in how vessels exposed to perturbations converge to distinct steady states, despite their nearly identical conditions.”

This response strongly implies true ecological multistability, defined as the coexistence of multiple stable states under the same external environment, differentiated only by their historical trajectories.

This narrative is consistent with the legend of Fig. 1:

“Fig. 1: Phenotype switching mechanism that leads to multistability. According to our hypothesis, in community contexts, multistability can arise from phenotype switching between subpopulations of the same species. This switching is triggered by small shifts in steady-state metabolite concentrations, which are influenced by environmental factors such as pH and dilution rate. Because the switching is not necessarily reversible

in a symmetrical manner, the system may exhibit history-dependent steady states (i.e., hysteresis).”

Nevertheless, the discussion section of the manuscript presents a subtly different scenario, suggesting that alternative states primarily arise from shifts in environmental conditions and phenotype responses, rather than historical contingency alone. For example, the authors state:

“However, in our system, these shifts are triggered by changes in environmental parameters and the associated microbial phenotypes.”

Thus, the manuscript’s narrative inadvertently drifts from strict multistability toward something resembling environmental hysteresis—where environmental conditions (even subtle differences) lead to different outcomes—rather than purely historical contingency under identical conditions.

To resolve this conceptual mismatch clearly, the authors should explicitly clarify in the manuscript whether their observed multistability arises strictly from historical contingencies under truly identical external environmental conditions, or whether subtle environmental variation indeed plays a crucial role. Such explicit clarification would greatly strengthen the manuscript’s conceptual coherence and precision.

We edited the text in lines 73-75, 281-283, and 413-416 and the legends of Figures 1, 4, 5 and 6 to tighten the concept. Additionally, we edited Figure 1 to make the relationship between the classical stability surface and the environment driver clearer:

Concerning the statement in the discussion that shifts in our system are triggered by changes in environmental parameters and the associated microbial phenotypes, we have now clarified that we refer to transient changes in environmental parameters.

5. Parameter fitting

The post hoc RMSE sensitivity plots (Fig. S7) suggest that a subset of fitted parameters are locally well-constrained, with RMSE minima centered near the estimated values. However, many parameters show minimal or no sensitivity across a wide range (0.1x-10x), indicating they may be poorly informed by the data or structurally non-identifiable. This concern is amplified by the use of Powell's method, which is a local, derivative-free optimizer that can struggle in high-dimensional parameter spaces.

Additionally, several fitted parameters appear pinned at or near their upper or lower bounds (e.g., `ri_xi_mumax`, `ri_xi_pHopt`, `ri_xi_pHalpha` in `ri.tsv` on GitHub; and many values in Table 2 of Supplementary Text S1 that round to integers like 1.000, 2.999, or 10.000). While some of these parameters show low sensitivity in the RMSE analysis, this does not preclude the possibility that their values reflect optimization constraints rather than genuine biological signals.

Thus, the authors should explicitly acknowledge these parameter-fitting limitations within the main text. Moreover, they should provide the full set of parameter bounds

used, clearly justified biologically or computationally (e.g., as Table 3 in Supplementary Text S1). Finally, I recommend conducting a formal parameter identifiability analysis or employing regularization methods to explicitly assess and demonstrate model robustness. Without addressing these concerns, the biological interpretability and predictive power of the model parameters remain uncertain.

We added the limitations of the parameter fitting to the discussion. Parameter bounds are summarised in a file shared on GitHub (<https://github.com/danielriosgarza/hungerGamesModel/blob/main/files/params/allParamsFitted.tsv>) and also shown in the new notebooks (see below). The bounds were selected based on previous knowledge, reasonable biological feasibility, and warm-up rounds of the optimization. We agree that the model can be improved by applying methods that reduce model complexity, but given that these are not trivial, that the model in its current form is well supported by the available data and reproduces key findings and that we already included a simpler (phenomenological) model, we consider further model refinements to be a task for the future.

Reviewer #4 (Remarks on code availability):

The authors provide executable Python notebooks for reproducing the main theoretical figures, which I greatly appreciate. However, the parameter fitting pipeline is less transparent. While parameter outputs and value constraints are included in the params/ folder, no executable code or documentation is provided for the fitting procedure itself. This limits reproducibility of the core modeling results and makes it difficult to evaluate identifiability or optimization behavior.

We agree that the code session for parameter fitting was not well organized. We organized it and made available three executable notebooks that demonstrate our parameter fitting approach for each of the strains:

[hungerGamesModel/notebooks/fitParameter2Model_bh.ipynb at main · danielriosgarza/hungerGamesModel](https://github.com/danielriosgarza/hungerGamesModel/blob/main/notebooks/fitParameter2Model_bh.ipynb)

[hungerGamesModel/notebooks/fitParameter2Model_bt.ipynb at main · danielriosgarza/hungerGamesModel](https://github.com/danielriosgarza/hungerGamesModel/blob/main/notebooks/fitParameter2Model_bt.ipynb)

[hungerGamesModel/notebooks/fitParameter2Model_ri.ipynb at main · danielriosgarza/hungerGamesModel](https://github.com/danielriosgarza/hungerGamesModel/blob/main/notebooks/fitParameter2Model_ri.ipynb)

Point-by-point reply

Reviewers #5 and #6

From reading the back and forth between the authors and reviewer #4 I feel comfortable with the conceptual approach being used by the authors. I would note that resource concentrations and pH can be part of the feedback loop leading to multi stability, so the fact that these concentrations are different at the end of the experiment does not mean that the observation is not "true multi stability". However, I do feel that the experiments are not the cleanest demonstration of multi stability. In particular, perturbations can induce transitions between alternative stable states, but they can also lead to extinction of a species and the resulting final state is not necessarily a stable state, in that it is possible that the final state of the community can still be invaded by the species that went extinct during the perturbation. For example, in Figure 5C the authors show that a Ph followed by feed perturbation leads to three possible outcomes: 1) coexistence of all three species, 2) extinction of *B. theta* and coexistence of the other two, and 3) extinction of two species and survival of *B. hydro*. I would note that these are not necessarily stable states, since it is possible (likely?) that outcomes (2) and (3) correspond to community states in which one or both of the extinct species could invade, leading the community to end up at state (1). I feel that the authors should at least acknowledge this as a possibility.

We thank the reviewers for their positive assessment and for raising this interesting possibility. We have acknowledged this limitation in the discussion, where we added to the paragraph :

A limitation of our work is that we did not collect direct evidence of phenotype switching in the community, which requires stable isotope experiments and is a task for the future. Another limitation is that we did not test whether the steady states reached in our chemostat system are robust to invasion. It is possible that a species goes extinct during the transient phase, but that once the system reaches a steady state, the conditions might allow the extinct species to reestablish itself in the community if re-inoculated. This possibility does not arise in the dynamic model, where all three species are always present, even if some persist at very low density. However, it remains to be tested whether this is also the case in our experimental system.

Reviewer # 4

First, we thank the reviewer for taking the time to thoroughly evaluate our work. Below we address the four points raised in their pdf. While these points continue the discussion from our previous revision, we still disagree on certain interpretations.

1. Conceptual framing: sharp transitions vs. alternative steady states

The reviewer raises an interesting possibility; however, in our view, the evidence strongly supports the multistability hypothesis than the threshold-triggered-by-noise hypothesis. As noted previously, the random variation introduced during the perturbation (observed as noise or variability between vessels) is not greater than the variation present prior and post-perturbation. Therefore, the

statement that “without evidence that environments and populations were equivalent immediately after perturbation, it remains difficult to attribute divergence solely to population composition” is, in our opinion, not correct. We do have evidence that the environmental condition (dilution rate, composition of the supplied medium) were equivalent within the limits of experimental noise. The changes in the population density and in the metabolic concentration can therefore be attributed to a switch to an alternative, coexisting state in the whole system.

Moreover, greater experimental noise is observed under control conditions, yet replicate vessels converge to the same steady state. If the hypothesis of two separate attractors were correct, we would expect to observe some control vessels in alternative states—which we do not. We therefore cannot disregard this evidence in favor of an alternative hypothesis for which there is less support.

We also note that our system is an emulated chemostat where one would not expect two distinct attractors connected by historical contingency. By restoring the system to its previous dilution rate and feed, one would expect the system to return to the same, or nearly the same, steady state. We acknowledge that metabolic memory or other factors could represent interesting exceptions to this rule, but these are difficult to test and remain open questions for future work. Finally, we also acknowledge the limitation raised by reviewer #5: we cannot rule out the possibility of extinction during the transient phase, which would represent an exception to the alternative-state hypothesis. This limitation has been added to the Discussion.

2. Modeling supports environmental determinism

Contrary to the reviewer’s claim, the model does predict the coexistence between 2 attractors in the same conditions. The model—coincidentally similar in structure to those described by May and Scheffer—exhibits the occurrence of true hysteresis, triggered either by dilution or by pH. While we use these perturbations to drive the system toward alternative states (to remain consistent with what is experimentally feasible), it is straightforward to verify that different initial conditions would also lead to alternative states, as we discussed extensively in the previous review round. For example, varying only the initial population of *Blautia hydrogenotrophica* results in distinct outcomes:

Initial populations:

Blautia hydrogenotrophica: 500 cells/ul

Bacteroides thetaiotaomicron: 300 cells/ul

Roseburia intestinalis: 300 cells/ul

Initial populations:

Blautia hydrogenotrophica: 300 cells/uL

Bacteroides thetaiotaomicron: 300 cells/uL

Roseburia intestinalis: 300 cells/uL

3. Reframing and summary

Based on the discussion above, our prior responses, and our interactions with the other reviewers, we believe that we have sufficient support for the occurrence of coexisting alternative states in our system and decided not to reframe our findings. We find little or no evidence for tipping behavior or ecological memory in the absence of multistability, and therefore continue to interpret our results within the framework of multistability.

Paper summary

The paper titled “discovery of alternative stable states in a synthetic human gut microbial community” utilizes three common human gut bacterial species to form simplified communities and analyzes the community composition under various environmental conditions and perturbations. The work combines experiment data and ODE simulations and suggest multiple steady states can emerge from the same community. In particular, the ODE simulations were generated from an elaborate community modelling framework, as the authors tried to provide a full metabolic pathway of the community. Moreover, on the empirical side, the authors provide a nice exploratory dataset including both population and nutrient dynamics with good temporal resolutions. However, there are multiple fundamental issues that led us to doubt the suitability of publishing in *Nature Communications*, as described in the comments below:

General comments

1. The main claim “finding the alternative stable states” is ill-supported by the experimental data. Firstly, no statistics were used to support the notion of alternative stable states, and we cannot tell if a distribution is truly bimodal from just two samples (bioreactor vessels). From the graphs, it looks like a range of shifting community states in response to but even in the absence of a perturbation. Secondly, another issue is that the data are inconclusive as to whether those states are stable, as there was no statistical analysis to support the claim on stability. Moreover, our third main criticism is of conceptual nature: The authors are not finding “alternative” stable states because the community does not converge on multiple alternative compositions under the **exact same** environmental conditions. Changes in community in response to environmental changes are hardly a novel finding, as they directly follow from the idea that bacteria adjust their gene expression to environmental changes (e.g. following by the seminal work of Jacob and Monod). If the message of this manuscript amounts to “a community changes in response to a different environment” instead of finding alternative stable states seriously undermine the novelty of this manuscript. In other words, the fact that a shift in diet changes the microbiome’s composition is not novel.
2. The main text failed to highlight the key messages; instead, it is a series of descriptions of metabolic phenomena/behaviors and lacks clarity which statement is derived of modeling versus experimental results. A concise graphical abstract showcasing the most important metabolic

functions/mechanisms that are supposed to give rise to the predicted alternative steady states would help.

3. The quality presentation needs to be improved to be understandable to general microbiologist audiences. For instance, in Fig. 4, the pH was not in the legend, and populations are presented in a cumulative style with very thin columns, which makes understanding the absolute abundance literally impossible. It is also unclear why the panel on the left-hand side shows only such sparse sequencing data in comparison to the right-hand side panel. Similarly, the differences between experiments in Fig. 1 is not clear (graphically). In addition, the bioreactor system figure in Fig. 2C is misleading because the coexisting graph is purely computational, not empirical. Lastly, the meaning of columns in Fig. 3C is unclear but different from Fig. 3A and B (which is the dilution rate); are they observations from different time points?
4. The model fitting process is only described with one sentence at the end of supplementary document. This is a huge problem considering there are more than 100 parameters in the full model, where fitting values is a very challenging task. The authors mentioned the loss function and the optimization algorithm, but completely skipped explaining the fitting routine, such as which variable was fitted first and how the nutrient data was processed before fitting. The online code repository has the very same problem, that there are codes for moving average and splines but no codes for the optimization process. In addition, there is no confidence or credible intervals of the fitted values, not to mention that the overall structure of the supplementary is sloppy (e.g., sections are not properly indexed and hard to navigate).
5. Model design is a bit under-justified. The authors used the metabolic pathway supported by RNA-seq data as the backbone of the entire model. Nevertheless, this approach does not exclude the possibility of additional pathways being omitted, which is important considering the communities grew in a rich medium. Similarly, it seems the authors put no effort to reduce the number of variables in the model. For example, why do we need an inactive and a dead population? If there is a close to zero probability for those cells to reactivate, then the dead population is redundant. In summary, some sensitive analysis should be considered to show the findings are robust against adding or removing some metabolic pathways.

Specific comments

- Author contributions: it is unclear who did the experiments.
- The PCA in Fig.3 is hard to understand and a 3D figure did not help.
- Why there is only two bioreactor data being shown while the device can run more? Was there data being thrown away?
- Consider renaming some variables to improve readability. For instance, change μ_{max_c} to $\mu_{max,c}$, and pH_{opt} to pH_{opt} .
- Add figure and table numbers to the supplementary document.
- Parameter table in the supplementary document: make sure the values are in the same format. Some 10 have different numbers of digit after the decimal place
- Supplementary document page 5, before the figure with M_a and M_b : a paragraph is duplicated.

I appreciate the substantial effort the authors have made in improving the clarity and transparency of their work. Given the importance of the central claim regarding alternative stable states, I would like to make one more round of clarification, framed constructively, to ensure conceptual precision in how the results are interpreted.

1. Conceptual framing: sharp transitions vs. alternative steady states

The main concern remains that the paper currently overstates the strength of the evidence for the existence of alternative stable states, when the results are better interpreted as deterministic transitions between regimes with a single attractor per condition. To clarify: my concern is not about unintended differences in feed composition or flow rate between vessels, but rather about the possibility that small variations during the perturbation phase—such as in the effective dilution of cells and metabolites—could lead to correlated shifts in both nutrient levels and population structure. **These post-perturbation states may then drive divergence deterministically, without invoking multiple attractors.** Without evidence that environments and populations were equivalent immediately after perturbation, it remains difficult to attribute divergence solely to population composition.

Figure 1. Distinguishing environmental determinism from true bistability requires decoupling post-perturbation environments and populations. (a) Replicate trials diverge due to subtle variations in environments during perturbation, consistent with deterministic transitions shown in the authors' simulations. Each solid circle

represents the post-perturb condition (abiotic on the x-axis and biotic on the y-axis), each dotted circle combines the environment and population from different trials, and each arrow indicates the recovery trajectory towards steady states. (b) True alternative steady states are demonstrated when the same environmental condition supports multiple steady states, achieved by replacing the population of a reactor with the population from another reactor, which is the evidence not yet shown in the current study.

To help clarify this distinction, I include a schematic figure that contrasts (Fig. 1).

- Environmental determinism: A sharp, deterministic regime shift where different outcomes are observed only because environmental conditions differ slightly across experimental replicates.
- Bistability: A true multistability scenario, where multiple outcomes emerge under the same external conditions, implying the coexistence of multiple attractors.

In the current study, community divergence occurs following perturbations, during a phase in which nutrient concentrations and population dynamics react on similar time scales (as shown in Fig. 6 and Supplementary Fig. S7). This temporal overlap makes it difficult to disentangle whether divergence is driven by internal system dynamics or by subtle, correlated variations during perturbations. For instance, if one reactor experiences slightly more feed flow than intended during perturbation, then both cell populations and preexisting metabolites could be disproportionately diluted. Therefore, the authors **do not provide evidence that the divergent steady states are independent of metabolite carryover or memory**, and thus attributable solely to differences in population composition. While the results compellingly demonstrate hysteresis and ecological memory, they do not conclusively establish the existence of alternative stable states.

2. Modeling supports environmental determinism

The ecological simulations, by construction, cannot exhibit true spontaneous divergence from identical initial states. While the model includes phenotype switching and hysteresis, the number of attractors at any fixed parameter setting remains one. This further supports the view that the observed experimental divergence likely results from subtle variations during recovery, not multistability. Thus, the modeling does not demonstrate multistability either—but it does a good job of explaining state switching triggered by threshold-crossing (Fig. 1a).

While the authors cite May and Scheffer in support of their modeling framework, both of those studies demonstrate multistability explicitly by plotting trajectories from multiple initial conditions under fixed external parameters, showing convergence to distinct equilibria. In contrast, the current study presents **hysteresis arising from changing parameters**, but does not test whether multiple steady states coexist at any fixed parameter setting. As such, the model's behavior is consistent with threshold-driven transitions, but does not replicate the phase-space multistability demonstrated in those classical models.

3. Reframing

In my earlier comments, I suggested a re-inoculation experiment to disentangle their results from environmental determinism. Since the authors have expressed no intention of pursuing this, I believe the best path forward is a careful reframing of the manuscript's central claims to ensure scientific accuracy without diminishing its genuine contributions to microbiology.

Currently, the paper frames its results as a demonstration of alternative stable states. However, both the empirical data and the deterministic model fall short of establishing the coexistence of multiple attractors under identical external conditions. Instead, the results more clearly show: (i) sharp transitions in community composition, (ii) ecological hysteresis and memory following perturbations, and (iii) phenotype switching as a plausible mechanistic driver of these transitions. These are important and novel findings. They would be stronger and more rigorous if framed as evidence consistent with multistability, rather than as direct proof.

Suggested edits to manuscript text:

Replace:

“We demonstrate the existence of alternative stable states...”

with:

We observe sharp, history-dependent transitions in microbial community composition under controlled, repeatable conditions, consistent with multistability mechanisms. However, due to practical limitations in verifying perfectly identical post-perturbation conditions, our results more

conservatively demonstrate tipping behavior and ecological memory, which are necessary but not sufficient evidence for alternative stable states.

In the discussion, consider clarifying:

While our modeling and experiments exhibit hysteresis and persistent divergence following perturbations, we acknowledge that they do not definitively demonstrate the coexistence of multiple attractors under strictly identical external conditions.

These changes would preserve the impact of the findings while enhancing conceptual clarity and precision.

To better reflect the results and avoid overstatement, I suggest the title:

Phenotype switching and history-dependent transitions in a synthetic human gut microbial community

This title still highlights the central mechanisms explored: phenotype switching and ecological memory, without asserting alternative stable states that have not been rigorously demonstrated.

4. Summary

To summarize, the authors have made significant contributions toward understanding ecological memory and threshold-driven transitions in microbial communities. However, the **current framing goes far beyond what the data and model justify** regarding multistability. A framing adjustment would substantially strengthen the manuscript's clarity and scientific rigor. I hope these clarifications are helpful and will be taken in the spirit of constructive dialogue aimed at sharpening the paper's scientific message.